# Biogenesis of DNA-carrying extracellular vesicles by the dominant human gut methanogenic archaeon

Diana P. Baquero [1] ✉, Guillaume Borrel [2], Anastasia Gazi [3],
Camille Martin-Gallausiaux [2], Virginija Cvirkaite-Krupovic [1],
Pierre-Henri Commere[4], Nika Pende [2,5], Stéphane Tachon[6],
Anna Sartori-Rupp [6], Thibaut Douché [7], Mariette Matondo [7],
Simonetta Gribaldo [2] & Mart Krupovic [1] ✉

Extracellular vesicles (EVs) play many important roles in cells from all domains of life. Here, we characterize EVs produced by *Methanobrevibacter smithii*, the dominant methanogenic archaeon in the human gut, which contains a peptidoglycan cell wall. We show that *M. smithii* EVs are enriched in histones and diverse DNA repair proteins. Consistently, the EVs carry DNA and are strongly enriched in extrachromosomal circular DNA (eccDNA) molecules, which originate from excision of a 2.9-kb chromosomal fragment, and a proviral genome. The eccDNA encodes enzymes implicated in biosynthesis of cofactor $F_{420}$ and coenzyme M, two elements critical for methanogenesis. Furthermore, several of the most abundant EV proteins are implicated in methanogenesis. Cryo-electron tomography suggests that EVs are formed by budding from the cell membrane and are trapped under the peptidoglycan layer prior to liberation through disruptions in the cell wall. Our results reveal parallels with EV biogenesis in bacteria and suggest that *M. smithii* EVs have potential impact on methane production in the gut.

Extracellular vesicles (EVs) are membrane-bound particles secreted into the extracellular environment by cells from all domains of life. EVs range in size from 20 to 400 nm in diameter and originate from cellular membranes[1]. Although EVs were initially thought to carry cellular waste products, represent cell debris or artifacts of lipid aggregation[2], it is now recognized that they play multiple important roles in diverse biological processes, including horizontal gene transfer, transport of metabolites, cell-to-cell communication, biofilm formation, virulence and antiphage defense[3–8]. Thus, understanding the role of EVs across domains of life emerges as an important research direction.

Most of the research on EVs thus far has focused on eukaryotes and bacteria. However, a growing body of evidence suggests that phylogenetically diverse archaea also produce EVs[9]. In particular, EVs have been reported in hyperthermophilic archaea of the orders Sulfolobales, Thermococcales and Methanococcales as well as halophilic archaea of the order Halobacteriales[7,10–17]. Although the impact of EVs on archaeal communities remains to be fully understood, it is becoming increasingly clear that they play an active role in various ecosystems. The first archaeal EVs to be characterized in the thermoacidophile *Saccharolobus islandicus* (order Sulfolobales) were found to carry an antimicrobial protein named 'sulfolobicin', which selectively

[1]Institut Pasteur, Université Paris Cité, CNRS UMR6047, Archaeal Virology Unit, Paris, France. [2]Institut Pasteur, Université Paris Cité, Evolutionary Biology of the Microbial Cell, Paris, France. [3]Institut Pasteur, Ultrastructural Bio Imaging, UTechS, Université Paris Cité, Paris, France. [4]Institut Pasteur, Flow Cytometry Platform, Paris, France. [5]University of Vienna, Archaea Physiology and Biotechnology Group, Vienna, Austria. [6]Institut Pasteur, NanoImaging Core Facility, Centre de Ressources et Recherches Technologiques (C2RT), Paris, France. [7]Institut Pasteur, Université Paris Cité, CNRS UAR2024, Proteomics Platform, Mass Spectrometry for Biology, Paris, France. ✉e-mail: dp.baquero645@gmail.com; mart.krupovic@pasteur.fr

inhibits the growth of other Sulfolobales species[18]. In contrast, *Thermococcus prieurii*, a sulfur-reducing hyperthermophilic member of Thermococcales, secretes EVs packed with elemental sulfur, likely to prevent the accumulation of intracellular cytotoxic levels of this element[12]. Archaeal EVs might also play an important role in horizontal gene transfer and DNA encapsulation, which appears particularly critical in extreme geothermal and acidic environments. Indeed, EVs produced by *S. islandicus*, *Thermococcus* and *Halorubrum* species were shown to carry chromosomal and/or plasmid DNA[7,10,13,14,19,20]. For instance, EVs from *S. islandicus* REY15A can transfer the plasmid-borne *pyrEF* locus into a plasmid-free auxotrophic *S. islandicus* strain[19]. Finally, under nutrient limiting conditions, *S. islandicus* EVs were shown to serve as a carbon and nitrogen source, promoting microbial growth[19].

The mechanisms of EV biogenesis remain poorly understood, but it is evident that the structure and composition of the cell envelope play a central role in this process. In most bacteria, the cytoplasmic membrane is encased by a rigid peptidoglycan layer, which presents a barrier for EV release. Two major mechanisms of bacterial EV production have been proposed. The first mechanism is common to both monoderm (containing a single cytoplasmic membrane) and diderm (containing an inner and an outer membrane) bacteria and occurs through explosive lysis caused by phage infection, induction of prophages or action of autolysins[21,22]. Accordingly, EVs produced through the explosive lysis route (E-type EVs) are enriched in peptidoglycan-digesting enzymes[23,24]. The second mode of EV release occurs via membrane blebbing (B-type EVs). In diderm bacteria, the EVs are normally produced by blebbing of the outer membrane. Recent super-resolution microscopy analysis of EV biogenesis showed that EVs of monoderm bacteria can be also produced by budding/blebbing from the cytoplasmic membrane[25]. These EVs undergo a 'waiting' period whereby they are trapped between the membrane and peptidoglycan layer until local openings in the cell wall allow their release[25]. EVs produced via explosive lysis are larger and more variable in diameter compared to the EVs produced by blebbing from the cytoplasmic membrane[23,25].

The envelope of most archaeal cells consists of a cytoplasmic membrane covered by a paracrystalline protein surface (S-) layer[26]. Thus, all archaeal EVs characterized thus far are covered by the cellular S-layer, consistent with budding from the cytoplasmic membrane[18,19,27]. The molecular mechanisms of EV budding have been studied in thermoacidophilic and halophilic archaea, *S. islandicus* and *Haloferax volcanii*, respectively. In *S. islandicus*, similar to eukaryotes[28], EV budding is mediated by the ESCRT (endosomal complexes required for transport) system[19], a membrane remodeling machinery which plays a key role in cytokinesis[29–31]. By contrast, in halophilic archaea, which lack the ESCRT system, EV biogenesis was shown to depend on the small Ras-like GTPase[15]. Both ESCRT machinery components and the GTPase are strongly enriched in the corresponding EVs[15,19]. Notably, members of the order Methanobacteriales, which include the dominant archaea in the animal and human gastrointestinal tract (GIT)[32], have a peptidoglycan polymer surrounding the cytoplasmic membrane and lack the S-layer[26]. Whether gut methanogens with a rigid cell wall can produce EVs remains unknown. This question is of particular interest given that EVs produced by gut bacteria play an important role in regulating the intestinal microenvironment, modulating both inter-microbial and microbe-host interactions[33,34].

Here, we characterize the composition and biogenesis of EVs produced by *Methanobrevibacter smithii*, the dominant archaeon in the human GIT, accounting for over 90% of the gut archaeome[35–37]. *M. smithii* is a strict anaerobe which obtains energy by reducing carbon dioxide into methane using molecular hydrogen as an electron donor[38]. *M. smithii* cells have an ovococcoid morphology and are surrounded by a peptidoglycan polymer as a primary cell wall component. We demonstrate that *M. smithii* EVs carry fragments of chromosomal DNA and are enriched in proviral DNA and

extrachromosomal circular DNA molecules. Notably, the EV proteome is dominated by proteins involved in DNA metabolism and includes several proteins implicated in methanogenesis. Cryo-electron tomography (cryo-ET) analysis shows that *M. smithii* EVs accumulate in the periplasmic space before being released into the extracellular environment through local disruptions in the cell wall, revealing parallels with the mechanism of EV biogenesis by monoderm bacteria.

## Results

### Characterization of EVs from *Methanobrevibacter smithii*

To investigate the potential production of EVs in host-associated peptidoglycan-containing methanogens, we chose the type strain *Methanobrevibacter smithii* PS (ATCC 35061) as a model organism. In the absence of protocols for EV purification from peptidoglycan-containing archaea, we first established a protocol for obtaining high-purity EV preparations from *M. smithii*, a prerequisite for biochemical characterization (Fig. 1A). The EVs were isolated from exponentially growing *M. smithii* cultures to limit the contamination of the EV preparations with cell debris. All sub-cellular particles from filtered cell-free supernatants were collected by ultracentrifugation and subsequently loaded at the bottom of an iodixanol (OptiPrep™) floatation gradient in which particles containing lipids are expected to float up due to their lower density compared to proteinaceous particles (Fig. 1A). The gradient was fractionated, and particles in the collected fractions were washed and concentrated in phosphate buffered saline (PBS) solution (Fig. 1A). The SDS-PAGE (Fig. 1B) and transmission electron microscopy (TEM; Fig. 1C) analyses showed that the EVs of variable diameters were predominantly present in the gradient fractions with the densities between 1.11 and 1.13 g/mL.

The size distribution of purified EVs was estimated using NanoFCM (Nano FCM Inc., Xiamen, China), a flow cytometer specifically designed for the analysis of nano-sized particles such as EVs and viruses. The majority of particles (~97%) measured using NanoFCM had diameters ranging from 45 to 100 nm (median diameter of 65 nm; Fig. 1D), while a minor fraction (~3%) of the detected EVs had diameters between 100 and 180 nm, which are herein referred to as large-diameter EVs (Fig. 1D and Supplementary Fig. S1). Analysis of the EV preparations using CytoFLEX Nano (Beckman Colter) yielded similar results, with the majority (93.4%) of EVs having a diameter of ~46–86 nm (Supplementary Fig. S2A). By contrast, the mean diameter of *M. smithii* EVs determined using Nanoparticle Tracking Analysis (NTA) was 98 nm (Supplementary Fig. S2B), which is at odds with the estimates obtained by the flow cytometry-based instruments. However, it is known that NTA has reduced sensitivity for EVs smaller than ~50 nm and results in an overestimation of EV sizes, due to the strong decrease in the intensity of scattered light scaling with particle diameter, which causes the scattered light of very small particles to disappear below the background noise[39]. Given that the detection limit of both NanoFCM and CytoFLEX Nano is ~40 nm and the inherent limitations of NTA, we also determined the size of *M. smithii* EVs by measuring the relative diameters of 412 particles imaged by TEM (Fig. 1D inset). The manually measured median diameter of EVs was 32 nm (min = 17 nm, max = 67 nm), indicating that flow cytometry-based instruments did not detect the majority of EVs produced by *M. smithii* (Fig. 1D and Supplementary Fig. S2A). No particles larger than 67 nm in diameter were observed by TEM, likely due to their lower abundance compared to the small-diameter EVs (~30 nm) (Fig. 1C). Collectively, our results suggest that *M. smithii* EVs are spherical, display a wide variation in diameter (~20–180 nm), with the majority of particles having a diameter of ~30 nm.

### *M. smithii* EVs preferentially enclose a 2.9 kb circular DNA molecule

EVs produced by hyperthermophilic and halophilic archaea were previously shown to carry DNA[7,10,14,19]. To determine whether EVs from

*M. smithii* also carry genetic material, the purified EVs were treated with DNase I to eliminate any extracellular DNA, followed by DNA extraction. Sequencing of the DNA purified from the EVs using the Illumina platform produced reads that mapped to the entire *M. smithii* chromosome with an average sequencing depth of 15 × (Fig. 2A, B ii). The small size of the *M. smithii* EVs precludes the encapsulation of a continuous complete chromosome within distinct EVs. Instead, it is likely that overlapping genomic fragments of varying sizes are randomly enclosed within the EVs, as has been reported for EVs from hyperthermophilic archaea[7,10,19] as well as bacterial EVs[40].

Unexpectedly, two chromosomal regions displayed remarkably high sequencing depths (Fig. 2A, B i and iii). The first, with an average sequencing depth of 9106 ×, i.e., ~ 600 × higher than the overall average, spans a chromosomal region between nucleotide positions 76,619 and 79,634 (Fig. 2B i). The reads covering this region were assembled into a circular contig of 2970 bp, suggesting the excision and circularization of the corresponding chromosomal region. To verify this assumption, we designed primers amplifying either the chromosomal locus or the excised element. Polymerase chain reaction (PCR) analyses and subsequent sequencing confirmed the presence of "integrated" and "excised" forms in both cells and EVs (Fig. 2C and Supplementary Figs. S3–S5), indicating that this locus indeed undergoes excision and circularization.

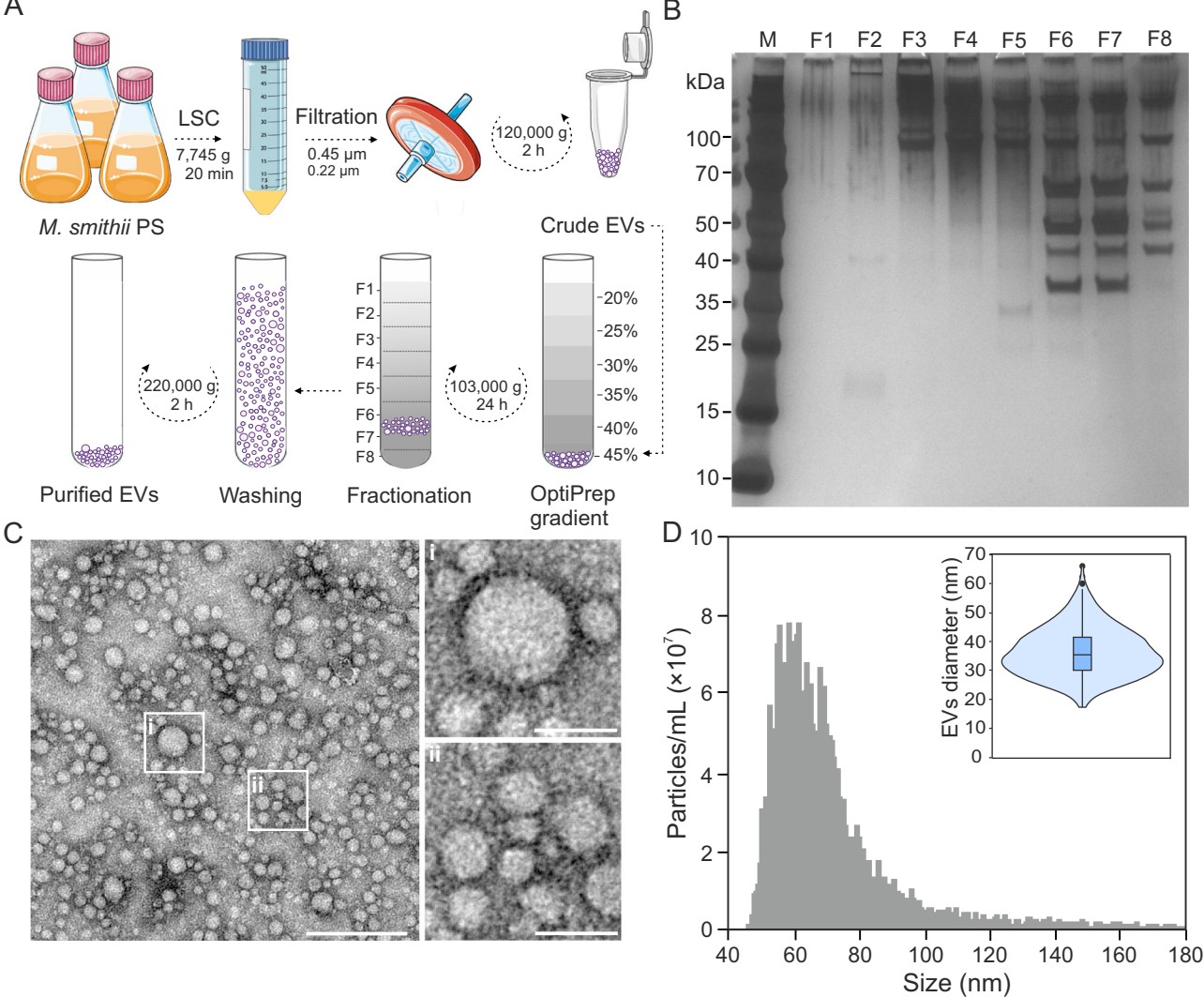

**Fig. 1 | Characterization of EVs produced by the methanogenic archaeon *Methanobrevibacter smithii* PS. A** Purification protocol established for *M. smithii* EVs. Crude EVs were obtained by ultracentrifugation of cell-free supernatants from *M. smithii* cultures. The pelleted crude EVs were resuspended in 45% OptiPrep™, and layers of lower concentrations were loaded on top. After ultracentrifugation of the OptiPrep gradient, fractions containing EVs were collected and combined. The collected EVs were washed in 10 times the volume of PBS solution and collected by ultracentrifugation. The pelleted purified EVs were then resuspended in PBS buffer. The schematic was partly created using graphical templates from Servier Medical Art under a Creative Commons license CC BY 4.0 (https://creativecommons.org/licenses/by/4.0/). The illustration was created with Corel Draw Graphics Suite 2021. **B** SDS-PAGE gel of fractions collected after OptiPrep gradient. Fractions 6 and 7 contain the largest number of EVs. **C** Transmission electron micrographs of *M. smithii* EVs. The main image displays EVs of different sizes. Scale bar, 200 nm. The upper right panel depicts a vesicle of approximately 60 nm in diameter. The bottom right panel displays smaller EVs with sizes ranging from 18–30 nm. Scale bar for both right panels: 50 nm. Samples were negatively stained with 2% uranyl acetate. **D** Particle size distribution determined by NanoFCM. The size distribution of *M. smithii* EVs was measured four times using different biological samples with NanoFCM (see Supplementary Fig. S1). The violin plot displays the range of EV diameters calculated manually using TEM imaging (*n* = 412). The width of the distribution indicates the frequency of occurrence. The samples used for the manual calculation were negatively stained with 2% uranyl acetate. The box plot displays the median (middle line), the 25th and 75th percentiles (box), the minimum and maximum values (whiskers), and the outliers (individual data points). Source data for Fig. 1D are provided as a Source Data file.

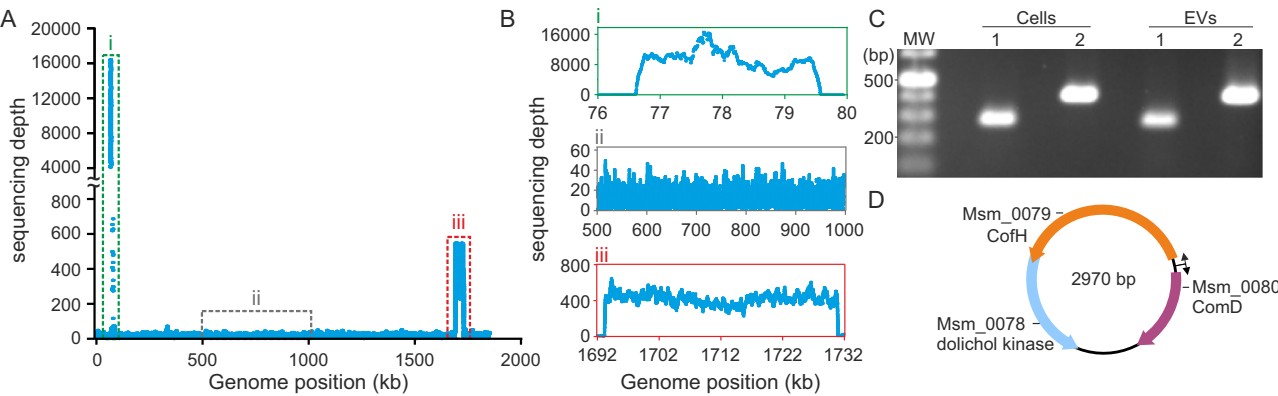

**Fig. 2 | DNA content of *M. smithii* EVs. A** Coverage depth across the *M. smithii* PS chromosome. Each dot represents the coverage at the indicated position. **B** Coverage depth across distinct regions of the *M. smithii* PS chromosome: region i (nucleotide coordinates 76–80 kb) giving rise to a circular element of 2.9 kb; region ii (nucleotide coordinates 500–1000 kb), representing the typical sequencing depth across the genome; region iii (nucleotide coordinates 1692 – 1732 kb), encompassing the MSTV1 provirus. Source data for Fig. 2A, B are provided as a Source Data file. **C** Detection of the integrated and excised form of the 2.9 kb DNA

fragment in both cell and purified EV preparations. The agarose gel electrophoresis shows the PCR amplified products: lane 1, integrated element within the host chromosome (expected size: 291 bp); lane 2, excised and circularized form of the 2.9 kb DNA element (expected size: 433 bp). **D** Genetic map of the circular element. The ORFs are represented by arrows indicating the direction of transcription. CofH, 5-amino-6-(D-ribitylamino)uracil:L-tyrosine 4-hydroxybenzyltransferase; ComD, sulfopyruvate decarboxylase subunit alpha. Genes *msm_0079* and *msm_0080* are preceded by divergent promoters, which are indicated with broken arrows.

The circular element carries three genes (*msm_0078–msm_0080*), none of which encode functions typical of mobile genetic elements (MGEs), i.e., integrases, transposases, or genome replication proteins. Instead, the three genes are implicated in different metabolic pathways (Fig. 2D). *Msm_0078* encodes a putative dolichol kinase (UniRef: A5UJA5), an enzyme involved in the synthesis of CDP-diglyceride, a compound that plays a key role in the biosynthesis of phosphoglycerides, one of the main structural components of biological membranes. The two other genes, *msm_0079* and *msm_0080*, encode 5-amino-6-(D-ribitylamino)uracil:L-tyrosine 4-hydroxybenzyltransferase (CofH; A5UJA6) and sulfopyruvate decarboxylase subunit alpha (ComD; A5UJA7), respectively. CofH catalyzes the production of 7,8-didemethyl-8-hydroxy-5-deazariboflavin, which is the precursor of the redox coenzyme $F_{420}$[38], and ComD is implicated in coenzyme M (CoM) biosynthesis[41], both critical for methanogenesis. In $CO_2$-reducing methanogens, cofactor $F_{420}$ is a crucial electron transporter, providing electrons from $H_2$ to reduce the methenyl group into methyl. In all types of methanogenesis pathways, CoM is the terminal methyl carrier before the formation of methane by the methyl-coenzyme M reductase complex[38].

Notably, the borders of excision of the circular element fall within protein-coding genes, truncating the genes *msm_0077* and *msm_0081* coding for thymidylate kinase (A5UJA4) and sulfopyruvate decarboxylase subunit beta (A5UJA8), respectively (Supplementary Fig. S3). The presence of putative promoters in front of *msm_0079* and *msm_0080* suggests that the two genes are transcribed, posing the intriguing question of the potential functions of such circular molecules in *M. smithii*.

## *M. smithii* EVs carry viral DNA

The second chromosomal region with a high sequencing depth of 420 × spans 38,824 bp (genome positions 1,693,231-1,732,055; Fig. 2A, B iii) and corresponds to a previously reported provirus in the genome of *M. smithii* PS[42]. We recently demonstrated that this provirus, named MSTV1, is sporadically reactivated in a small fraction of the *M. smithii* population, producing extracellular virus particles with a siphovirus-like morphology[43]. MSTV1 is present in 20% of all sequenced *M. smithii* strains and is likely to be the most abundant archaeal virus in the human gut[43]. PCR analyses revealed the excised form of the virus genome in both *M. smithii* cells and EVs (Supplementary Fig. S6). The

excision takes place at the proviral attachment sites and hence is mediated by the virus-encoded site-specific integrase of the tyrosine recombinase superfamily. The high sequencing depth of the provirus compared to the flanking regions (420× versus 15×) suggests that this locus in EVs is represented not only by the randomly incorporated fragments of the chromosomal DNA (as the rest of the chromosome) but also the actively excised viral DNA. The larger EVs observed by nanoFCM (Fig. 1D) have sufficient internal volume to accommodate the complete viral genome and could facilitate virus spread to non-infected cells in the population. Unfortunately, this hypothesis could not be confirmed experimentally using the *M. smithii* strains available in the laboratory, either due to the inability of the EVs to overcome the cell wall barrier of the target cells or due to resistance mechanisms that remain to be understood.

## *M. smithii* EVs are enriched in DNA-binding and DNA repair proteins

To assess the protein composition of the *M. smithii* EVs, we performed quantitative proteomics analysis of the *M. smithii* cells and EVs using mass spectrometry. The analysis led to the identification of 1073 proteins in cells and 417 proteins in EVs, which were present in all three biological replicates (Supplementary Fig. S7). None of the proteins were exclusive to the EVs (Supplementary Data 1). The number of proteins found in *M. smithii* EVs is similar to that described in EVs produced by other archaea, such as the hyperthermophilic archaeon *S. islandicus* (413 proteins) and the halophilic archaeon *Halorubrum lacusprofundi* (447 proteins)[14,19].

Archaeal Clusters of Orthologous Groups (arCOG)[44] were used to classify the potential functions of the *M. smithii* EV proteins (Supplementary Data 2). Proteins from the arCOG category P (Inorganic ion transport and metabolism) were found exclusively in the cellular proteome (Fig. 3A). In contrast, EVs were more enriched in proteins of the arCOG category J (Translation, ribosomal structure, and biogenesis), comprising 19% of the total EV proteins, compared to 13% in the *M. smithii* cell proteome. EVs also exhibited enrichment in proteins from the arCOG categories C (Energy production and conversion), E (Amino acid transport and metabolism), F (Nucleotide transport and metabolism), G (Carbohydrate transport and metabolism), and O (Posttranslational modification, protein turnover, chaperones) (Fig. 3A). The presence of proteins from nearly all arCOG categories in

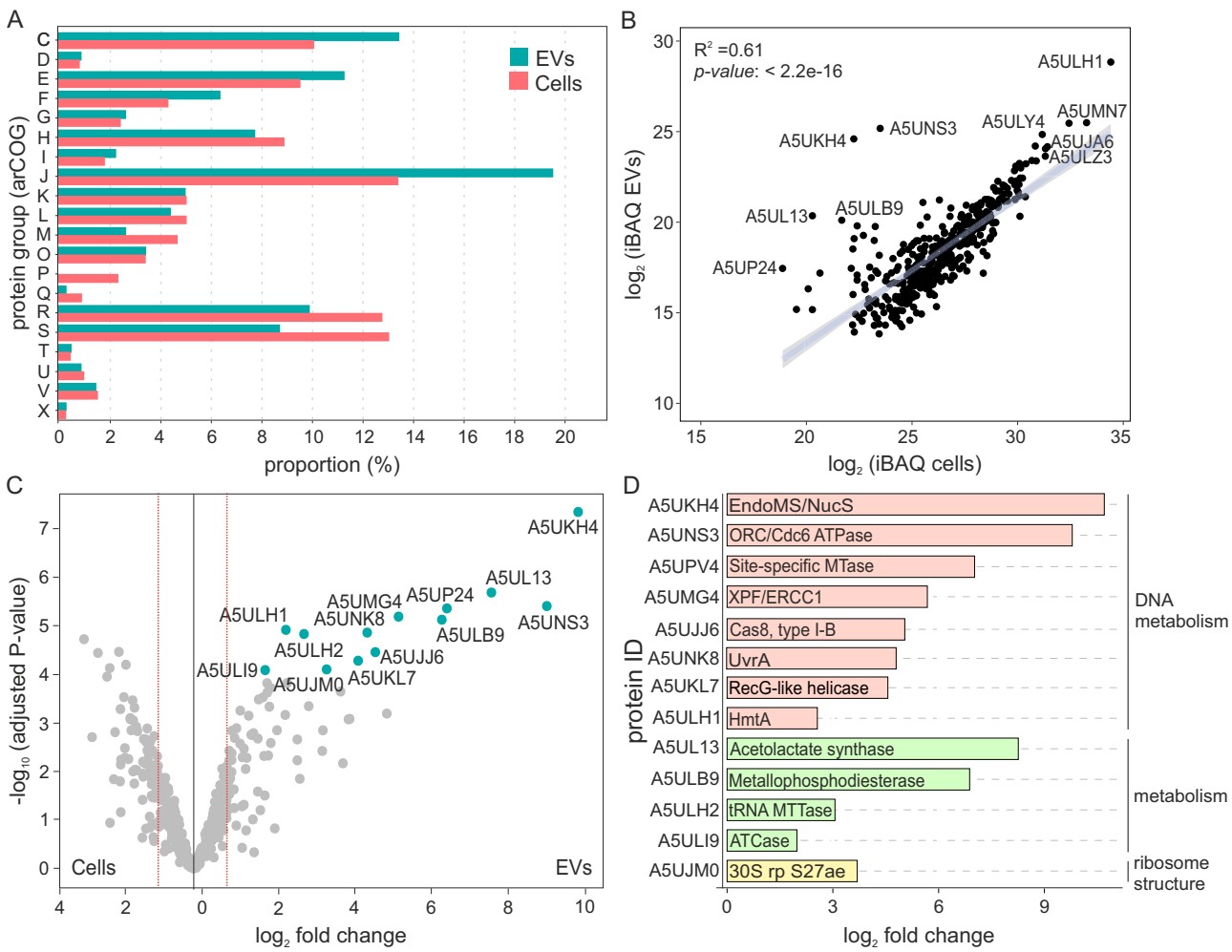

**Fig. 3 | Analysis of protein content of *M. smithii* EVs. A** Functional classification of proteins identified in purified *M. smithii* EVs using archaeal clusters of orthologous groups (arCOGs). Annotations for the arCOG categories are provided in Supplementary Data 2. **B** Correlation between protein abundances in *M. smithii* EVs and cells. Simple linear regression was performed, and the best fit line is shown (intercept = − 2.72, slope = 0.80, *p-value* < 2.2e-16, R² = 0.6004). **C** Volcano plot comparing protein content of *M. smithii* EVs and cells. Read lines highlight the

threshold for enrichment (*p* = 0.05 and |fold change| = 1.5). Differential protein abundances and adjusted *p*-values were calculated using the package DEP (Differential Enrichment analysis of Proteomics data). Blue dots indicate individual proteins significantly enriched in EVs. **D** Proteins significantly enriched in EVs when compared to proteins in the cellular fraction. The source data for Fig. 3 are provided as a Source Data file.

the EVs suggests that most of the proteins are incorporated nonselectively, likely by entrapment of the cytosolic and membrane contents, as previously suggested for other EVs[19]. Indeed, there is a strong positive correlation between the abundance of the proteins in the cells and their abundance in EVs (Pearson correlation coefficient r = 0.9189; Fig. 3B).

Label-free intensity-based absolute quantification (iBAQ) of the *M. smithii* EV proteome showed that the three most abundant proteins (Supplementary Data 3) are the three histone paralogs encoded by *M. smithii* (A5ULH1, A5UJP0, A5UMN7)[45]. Histone-fold proteins have been shown to mediate genome compaction in the order Methanobacteriales[46,47]. These observations suggest that the DNA in the EVs is in the form of DNA-histone complexes. Notably, the top four and top six overall most abundant proteins in the EVs are also associated with DNA metabolism (Supplementary Data 3). The top four most abundant protein is a homolog of the replication initiation Orc1/Cdc6 AAA + ATPase (A5UNS3) encoded by the MSTV1 provirus (Fig. 2B and Supplementary Data 3) and predicted to be involved in a regulatory circuit controlling the switch between the temperate and lytic states of MSTV1[43]. Notably, Orc1/Cdc6 is a non-structural protein (i.e., not part of the viral particles) and is the only provirus protein found in

all three replicates of the EVs proteome, confirming that the viral DNA detected in the EVs does not originate from contaminating virus particles. Given that Orc1/Cdc6-like proteins function through binding to DNA, it is likely that this protein is incorporated into EVs along with the viral DNA. This result is consistent with the EV purification strategy used, where proteinaceous virus particles are not expected to co-float with the lipid-containing membrane vesicles. Indeed, the density of *M. smithii* EVs is considerably lower than that of tailed virus particles (1.11–1.13 versus >1.2 g/mL)[48], and no virus particles were observed in the EV preparations by TEM. The top six most abundant protein in the EVs is the endonuclease EndoMS/NucS (A5UKH4) (Supplementary Data 3), a multifunctional enzyme involved in DNA repair processes such as nucleotide excision repair, mismatch repair, and deaminated base repair[49,50]. Presumably, EndoMS/NucS is incorporated into EVs along with its damaged DNA substrates.

Notably, four of the top ten most abundant EV proteins are related to methanogenesis (Supplementary Data 3). In particular, N⁵-methyl-tetrahydromethanopterin:coenzyme M methyltransferase subunit H (MtrH), the above mentioned 5-amino-6-(D-ribitylamino)uracil:L-tyrosine 4-hydroxyphenyl transferase CofH involved in $F_{420}$ biosynthesis, methyl-coenzyme M reductase subunit gamma, and $F_{420}$-dependent

methylenetetrahydromethanopterin dehydrogenase (A5ULY4, A5UJA6, A5ULZ3, and A5UMI1, respectively). CofH is the fourth most abundant protein in the total *M. smithii* proteome (Supplementary Data 4). It is tempting to speculate that the high expression of this protein could be at least partly caused by the extra gene copies borne on the excised circular DNA molecules.

Another notable protein present in high abundance in the EVs (top 14) is an adhesin (Msm_1398; A5UN25) with an N-terminal pectin lyase domain and four tandem immunoglobulin (Ig)-like domains at the C-terminus (Supplementary Data 3). Adhesins have been suggested to play a key role in ensuring the persistence of *M. smithii* PS in the distal intestine[51]. Similarly, Ig-like domains are specifically enriched in gut-associated archaeal viruses and are thought to mediate adhesion of virus particles to various substrates, including cell surface exposed glycans and the eukaryotic mucus layer[52]. The adhesins present in EVs could play a similar role. Notably, the same protein is also one of the most abundant adhesins in the total *M. smithii* proteome (top 11; Supplementary Data 4).

Next, we computed the differential protein abundance in the EVs compared to the cellular proteome (log2 > 1.5, adjusted *P*-value < 0.05) (Fig. 3C). This analysis identified 13 proteins that are significantly enriched in EVs as compared to the total cell proteome (Supplementary Data 5). Besides the endonuclease EndoMS/NucS (A5UKH4), one of the histone paralogs (A5ULH1), and the viral Orc1/Cdc6 AAA+ ATPase (A5UNS3), several other DNA metabolism and repair proteins were found to be among the most significantly enriched proteins in *M. smithii* EVs when compared to the cell proteome (Fig. 3B–D). These include the UvrABC nucleotide excision repair system subunit UvrA (A5UNK8), the nucleotide excision repair XPF/ERCC1 family helicase-nuclease (A5UMG4), the RecG-like ATP-dependent DNA helicase (A5UKL7), the adenine-specific DNA methyltransferase (A5UP24), and the type I-B CRISPR-associated protein Cas8 (A5UJJ6). The high abundance of proteins related to chromatin organization and DNA metabolism in the EVs suggests that the DNA fragments covering the entire *M. smithii* chromosome could be generated during the DNA repair processes, in particular, the nucleotide excision repair pathway, which generates fragments of damaged DNA[53].

Notably, the *M. smithii* genome does not encode orthologs of the ESCRT system or small Ras-like GTPase, which were shown to mediate EV budding in other archaea[15,19]. The only two GTPases (A5ULJ6 and A5UL42) identified in the *M. smithii* EV preparations were neither enriched nor abundant in the EVs, ranking 377 and 381, respectively (Supplementary Data 1). Thus, it is unlikely that EV biogenesis in peptidoglycan-containing archaea is dependent on these two GTPases.

### Cryo-ET provides insights into EV production in *M. smithii*

To gain insights into the biogenesis of *M. smithii* EVs, we analyzed the exponentially growing *M. smithii* cells by cryo-electron tomography (cryo-ET). Spherical EV-like particles (*n* = 31) were visualized within the cells, with diameters ranging from 12 to 45 nm (median diameter of 22 nm). The difference between the median EV diameters obtained by negative staining (32 nm) (Fig. 1D) and cryo-ET is likely due to artifacts inherent to the staining process. EVs were typically trapped in 'pockets' between the cytoplasmic membrane and the archaeal peptidoglycan polymer (periplasmic space) (Fig. 4A and Supplementary Movie S1). The reconstructed tomograms also revealed potential extracellular EVs in close proximity to the imaged cells. Both the periplasmic and extracellular EVs lack an apparent peptidoglycan coat.

Our data provides possible clues on how the EVs pass through the peptidoglycan polymer. We observed an area in the vesiculating cell that corresponds to a local opening (22 nm in diameter) in the cell wall, which colocalizes with the outward protrusion of the cell membrane and the presence of EVs outside of the cell (Fig. 4A). While it seems plausible that *M. smithii* EVs traverse through these areas, the mechanism of archaeal cell wall degradation and the components

involved in this process are unknown. We note that the cell, which was captured in the process of EV biogenesis, is undergoing division (Fig. 4A and Supplementary Movie S2), a time point in the cell cycle during which the cell wall is subjected to active remodeling. Notably, proteomic analysis has shown that EVs contain all four PRC-barrel proteins (arCOG02155), Msm_1004 (top 20), Msm_0841 (tope 43), Msm_0465 (top 56) and Msm_0822 (top 104), recently shown to be involved in cell division in *Haloferax volcanii*[54,55], as well as FtsZ (Msm_0626; top 52) (Supplementary Data 3), but not SepF (Msm_0406), which was detected exclusively in the cellular proteome (Supplementary Data 4).

In monoderm model bacteria, such as *Bacillus subtilis*, which harbor a thick cell wall of 20–40 nm, the action of phage- or host-encoded peptidoglycan-degrading enzymes has been shown to facilitate the transit across the cell wall[56–58]. Consistently, the EVs in these bacteria typically carry either phage-encoded lysins or cellular peptidoglycan-digesting autolysins[59,60]. In contrast, the PeiW-like endolysin (Peptidase_C71 family) encoded by the *M. smithii* provirus MSTV1[52] or any other putative (auto)lysins were not detected in EVs by proteomics analysis. Thus, the mechanism of EV production in *M. smithii* could be either (auto)lysin-independent, or the enzymes implicated in peptidoglycan remodeling could be distinct from those previously characterized.

## Discussion

The EVs are implicated in a variety of important processes, including intercellular communication, horizontal gene transfer, defense against viruses and more, but until now remained uncharacterized in methanogenic archaea. Here, we studied the EVs from the peptidoglycan-containing methanogen *Methanobrevibacter smithii*, the dominant archaeon in the human gut. We show that *M. smithii* EVs carry three types of DNA molecules, which vastly differ in their abundances. These correspond to (i) random genomic loci distributed across the *M. smithii* chromosome (15× depth), (ii) the MSTV1 provirus (420× depth) and (iii) an extrachromosomal circular DNA (eccDNA) molecule of 2.9 kb (9106× depth). The frequency of incorporation into EVs likely depends on the availability (copy number) and dimensions of the corresponding molecules in the cell during EV biogenesis. The fragments cumulatively covering the entire chromosome may be generated during various DNA repair processes, including DNA mismatch and nucleotide excision repair (NER) pathways. Indeed, EndoMS/NucS, the top six most abundant protein in EVs, is the major player in mismatch repair in most archaeal lineages[50,61,62]. By contrast, UvrA and XPF proteins, which are significantly enriched in EVs, are key players in the NER pathway, with UvrA being responsible for recognition of DNA helix-distorting lesions and XPF implicated in nicking the damaged DNA strand[53]. In bacteria, two different nucleases, XPF and XPG, nick the damaged DNA downstream and upstream of the lesion, respectively. However, since archaea lack homologs of XPG, it was suggested that XPF could cut the DNA on both sides of the lesion[53]. Alternatively, it has been proposed that EndoMS/NucS participates in the NER pathway together with XPF[49]. Thus, small linear DNA fragments generated through the NER pathway might be enclosed within the EVs. Due to the inherently random distribution of mutations in the population, different cells would be expected to produce EVs with random genomic fragments. Notably, the two high-frequency regions (the provirus and eccDNA) are also incorporated into EVs as part of the chromosomal DNA (i.e., in their integrated form), presumably with the same frequency as the flanking regions, as confirmed by PCR analysis with the primers amplifying across the integration borders. However, the presence of the provirus and eccDNA in the extrachromosomal form increases their copy number and hence the probability of being incorporated into EVs, partly explaining their higher abundance in the EVs. Although we did not detect specific sequence features, such as direct repeats, which would suggest site-specific excision of the 2.9 kb eccDNA

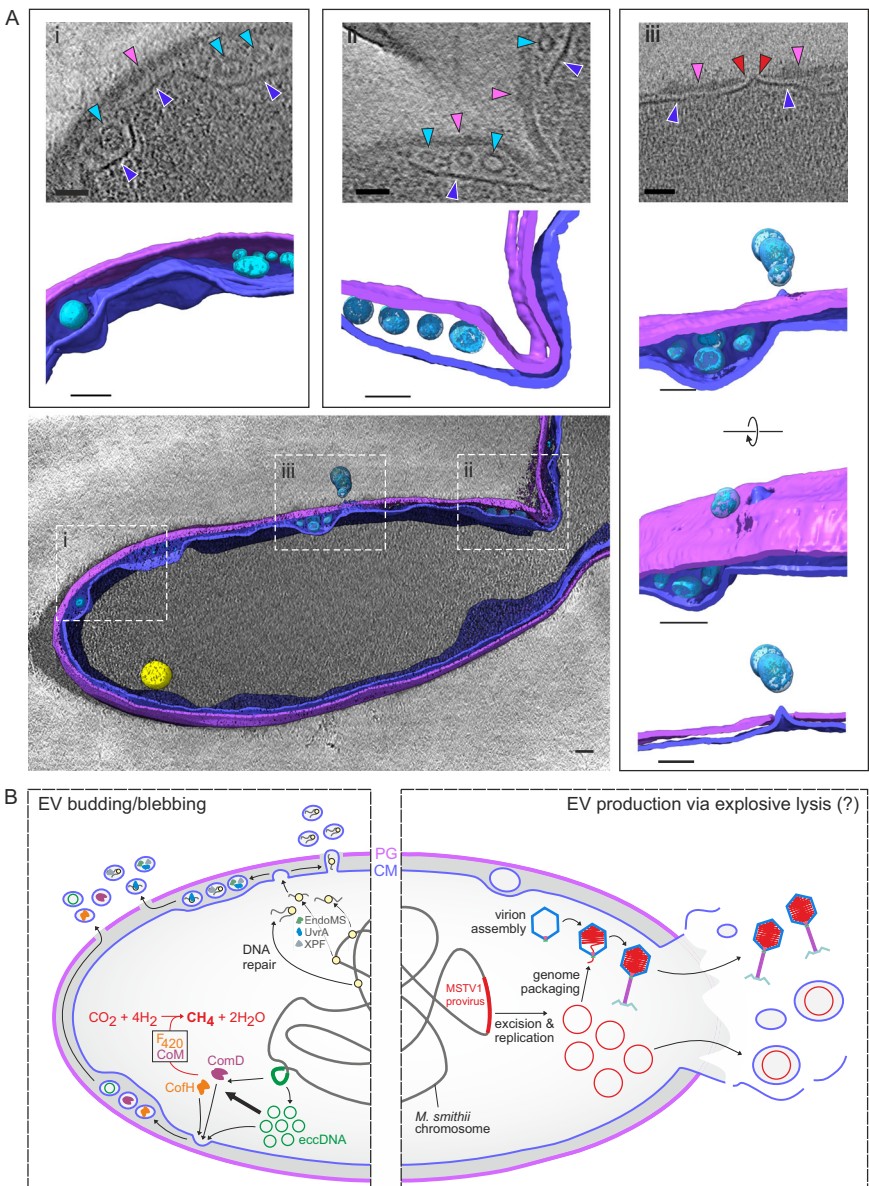

**Fig. 4 | Biogenesis of extracellular vesicles (EVs) by *M. smithii* cells.**
**A** Tomographic reconstruction of an *M. smithii* cell producing EVs. The central panel shows a segmented, surface-rendering display of the reconstructed tomogram of an *M. smithii* cell producing EVs. The reconstruction displays the following cellular components: membrane (purple blue), cell wall (purple), EVs (light blue), storage granule-like structure (yellow). Panels i and ii exhibit EVs trapped in 'pockets' between the cytoplasmic membrane and the archaeal peptidoglycan. Panel iii displays different tilts of a region in the cell where the peptidoglycan is locally disrupted, with the opening coinciding with the protrusion of the cell membrane and the presence of EVs outside of the cell. The upper images in panels i, ii and iii display distinct slices from the reconstructed tomogram. Colored arrowheads point to different structural components: light blue for EVs, blue for the cytoplasmic membrane, pink for the peptidoglycan, and red for local openings in

the cell wall. Scale bars, 50 nm. Two *M. smithii* cells were visualized producing EVs. **B** A schematic representation of EVs biogenesis and release by an *M. smithii* cell. DNA fragments from random chromosomal loci are incorporated into the *M. smithii* EVs. Two chromosomal regions, corresponding to the MSTV1 provirus and a 2.9 kb extrachromosomal circular DNA (eccDNA) molecule, are enriched in the *M. smithii* EVs. Consistent with the presence of DNA, proteins responsible for chromatin structure and DNA repair are also enriched in *M. smithii* EVs. Our cryo-ET results show that small EVs of approximately 12–45 nm in diameter are trapped in 'pockets' between the cytoplasmic membrane (CM) and the peptidoglycan (PG). We hypothesize that the release of small (~32 nm) and large (>100 nm) EVs in *M. smithii* occurs via distinct routes, budding/blebbing (left) and explosive virus-mediated lysis (right), respectively. The illustration was created with CorelDRAW Graphics Suite 2021.

---

segment, we cannot rule out the possibility that this chromosomal region is preferentially incorporated into EVs due to the importance of the encompassed genes for cellular fitness.

The cell envelope architecture of *M. smithii* could be another important factor underlying the incorporation frequency of different cargo DNA molecules. The size of the EVs is likely to be dictated, at least in part, by the diameter of the (non-lethal) openings in the peptidoglycan cell wall. Accordingly, EVs with diameters small enough to

pass through the peptidoglycan holes are expected to enclose only DNA molecules of relatively small size. The median diameter of the *M. smithii* EVs measured from TEM images is ~32 nm, consistent with the EVs observed by cryo-ET. Note that the diameter of the MSTV1 capsid in which the 38 kb viral genome is packaged under pressure is ~65 nm[43]. Thus, although histone proteins detected in the EVs are likely to condense the cargo DNA, the volume and hence the diameter of the EVs enclosing the viral DNA is still likely to exceed 65 nm. NanoFCM

analysis revealed a minor fraction (3.3%) of larger-diameter EVs ranging from 100 and 180 nm, which could be sufficient for carrying the full viral genome. Although it is conceivable that the EVs could be hijacked by MSTV1 to provide an alternative route for virus spread in the population, it is also possible that viral genomes are excreted from the cells as an innate antiviral defense strategy. Notably, one of the most abundant proteins in the EVs corresponds to the viral Orc1/Cdc6 AAA + ATPase (A5UNS3), an enzyme thought to be involved in the switch from the temperate to the lytic state of MSTV1[43]. Export of this protein from the cell may ensure stable lysogeny, and, therefore, host survival.

Random chromosomal fragments, plasmids and viral DNA were previously detected in EVs produced by halophilic and hyperthermophilic archaea[10,14,19,20]. By contrast, the small eccDNA molecules encoding metabolic genes described here were previously not reported in archaea. These molecules resemble the eccDNA described in diverse eukaryotes, including plants, nematodes, ciliates, yeast, and mammals[63]. eccDNA have been known in eukaryotes for over five decades, yet their significance and mechanisms of biogenesis remain enigmatic. Different strategies for the formation of circular extrachromosomal elements have been proposed, such as spurious homologous recombination between tandem repeats[64], or DNA damage repair processes and genome rearrangements caused by transposable elements[65]. Additionally, polymerase slippage in regions with repetitive sequences can create DNA loops during replication or repair that, when excised, result in the extrachromosomal circular elements[63,66]. We hypothesize that the 2.9 kb circular DNA element found both in *M. smithii* cells and EVs is excised from the chromosome by a similar, yet currently not understood, mechanism. In the case of the *M. smithii* eccDNA, two of the three genes encode enzymes implicated in methanogenesis, with one of these proteins being the fourth and eight most abundant protein in the *M. smithii* and EV proteomes, respectively. In addition, three other methanogenesis-related enzymes are among the top ten most abundant EV proteins. Thus, EVs could provide the means for efficient and rapid discharge of the extra gene copies and proteins once they are no longer needed in high quantities.

In hyperthermophilic archaea, EVs provide means for horizontal gene transfer. However, given that Methanobacteriales cells are surrounded by a rigid cell wall, fusion of EVs with the cytoplasmic membrane might not be straightforward, if at all possible. It is thus more likely that EVs provide means for the discharge of damaged or viral DNA, or surplus components, both proteins and genes, that outlived their usefulness under given conditions. Notably, the liberation of EVs trapped in the periplasmic space might not be the only possible outcome. Conceivably, the EVs could fuse back to the cytoplasmic membrane, reintroducing their cargo (e.g., eccDNA) into the cell. Under this scenario, the *M. smithii* EVs would serve an important regulatory role, fine-tuning methanogenesis to changing environmental or cellular cues. Timely reaction and adaptation to environmental changes might be particularly important in the gut context, where methanogenic archaea represent a minor fraction of the microbial community and face strong competition with bacteria.

The mechanism of EV biogenesis in prokaryotes is an outstanding question. Our cryo-ET analysis shows that the small-diameter *M. smithii* EVs (~32 nm) are present in the periplasmic space prior to their release into the extracellular milieu. This observation is inconsistent with blebbing through the holes in the peptidoglycan layer due to high turgor pressure, one of the models suggested for some bacterial EVs[67]. All archaeal EVs characterized thus far have been shown to be covered by the cellular S-layer[18,19,27], strongly supporting a mechanism in which EVs are formed by budding from the cytoplasmic membrane. However, the visualization of small EVs trapped in the periplasmic space of an *M. smithii* cell (Fig. 4A), along with the apparent absence of a peptidoglycan polymer surrounding *M. smithii* EVs suggests that the

biogenesis of EVs in Methanobacteriales follows a mechanism distinct from that proposed for other archaeal EVs. The process appears to be similar to that described in a recent super-resolution microscopy analysis of EV biogenesis in *Staphylococcus* species, whereby EVs undergo a 'waiting' period in the periplasm until local openings in the peptidoglycan layer allow their release (Fig. 4B)[25]. Furthermore, periplasmic EVs have been also observed in diderm bacteria but only when specific cell envelope genes, such as *tolB*, *nlpI* and *mlaE*, have been knocked out[68,69].

The release of larger-diameter EVs (>100 nm) carrying the viral genome would necessitate large openings in the peptidoglycan cell wall that could have detrimental effects on cell integrity. Therefore, the mechanisms generating small and larger EVs could be distinct. Bacterial EVs carrying viral genomes are thought to be generated during explosive virus-mediated lysis. Although we did not observe explosive lysis and production of larger EVs in cryo-ET, we hypothesize that the *M. smithii* EVs containing the MSTV1 genome are generated through a similar process (Fig. 4B). However, this hypothesis remains to be validated experimentally. More generally, the mechanisms of EV biogenesis in archaea appear to be more complex and diverse than previously appreciated, with at least three distinct proposed EVs biogenesis pathways. In particular, EVs of thermoacidophilic archaea of the order Sulfolobales are generated by budding in an ESCRT-dependent manner[19]; in halophilic archaea that lack the ESCRT system, EV budding depends on a small Ras-like GTPase[15]; by contrast, EVs of peptidoglycan-containing methanogenic archaea, which lack both the ESCRT machinery and the Ras-like GTPase, appear to be produced by blebbing, likely due to turgor pressure. The first two pathways are similar to those operating in eukaryotes, whereas the third one resembles the vesiculation mechanism reported in bacteria. Thus, biogenesis of archaeal EVs seems to occur through a combination of different mechanisms operating in the two other cellular domains and depends on the availability of molecular membrane remodeling machineries and the diversity of the cell envelope architecture. Whether cell division proteins detected in EVs, namely, the four PRC-barrel proteins and FtsZ, are involved in vesiculation remains to be explored experimentally once genetic tools for *M. smithii* become available.

Taken together, our results uncover similarities between EV biogenesis in bacteria and peptidoglycan-containing archaea and suggest that *M. smithii* EVs facilitate the export of DNA and metabolic proteins, especially those involved in methanogenesis, with potential impact on methane production in the gut environment. It is increasingly recognized that gut bacteria-derived EVs modulate the intestinal microenvironment and play an important role in gut homeostasis[33]. Although the role of archaeal EVs in the human gut remains to be determined, the fact that *M. smithii* EVs transport both DNA and proteins suggests that they may act as messengers or regulators in archaea-archaea, archaea-bacteria and archaea-host interactions. Our results could lay a foundation for harnessing the EVs produced by *M. smithii* for developing drug and vaccine delivery systems in the human gut[70].

## Methods

### *M. smithii* growth conditions

*Methanobrevibacter smithii* PS (ATCC 35061/DSM 861) cultures were grown at 37 °C, with agitation at 140 rpm, in serum bottles under strict anaerobic conditions (the gas phase comprised 80% $H_2$ and 20% $CO_2$ at 2.0 bar) in modified DSM 119 Methanobacterium medium containing 0.5 g/L $KH_2PO_4$, 0.4 g/L $MgSO_4$ x $7H_2O$, 0.4 g/L NaCl, 0.4 g/L $NH_4Cl$, 0.05 g/L $CaCl_2$ x $2H_2O$, 2 mg/L $FeSO_4$ x $7H_2O$, 1 mL trace element solution SL-10 (from DSM 320 medium), 1 g/L yeast extract, 1 g/L Na-acetate, 2 g/L Na-formate, 1 mL Selenite-tungstate solution (0.40 g/L NaOH, 6.00 mg/L $Na_2SeO_3$ x 5 $H_2O$, 8.00 mg/L $Na_2WO_4$ x 2 $H_2O$), 0.5 g/L tryptone, 0.5 mL/L Na-resazurin solution 0.1% w/v, 4 g/L $NaHCO_3$, 0.5 g/L L-Cysteine-HCl, 0.5 g/L $Na_2S$ x $9H_2O$, and 10 mL vitamin

solution (from DSM 141 medium). The medium was prepared as described previously[71], and the pH was adjusted to 7 with HCl.

## Isolation and purification of EVs

5 mL of an exponentially growing culture of *M. smithii* PS were inoculated into 45 mL of the modified 119 Methanobacterium medium and grown at 37 °C as described above. When cultures reached $OD_{600}$ ~ 0.35, cells were diluted into fresh modified 119 Methanobacterium medium with an initial $OD_{600}$ of 0.05, grown for 5–8 days and periodically gassed with $H_2$ and $CO_2$, maintaining the ratio 80:20 until they reach an $OD_{600}$ ~ 0.30–0.35. Then, cells were removed by low-speed centrifugation (Eppendorf F-35-6-30 rotor, $7745 \times g$, 20 min, 20 °C), supernatant was filtered through 0.45 and 0.22 μm filters (Merck Millipore) and ultracentrifuged to pellet the EVs ($120,000 \times g$, 2 h, 15 °C, Beckman 45 Ti rotor). After the run, the supernatant was removed, and the pellet was resuspended in 45% Opti-prep™. Different density gradient solutions (20%, 25%, 30%, 35%, 40% and 45% Opti-prep™ with calculated densities 1.11 g/mL, 1.137 g/mL, 1.163 g/mL, 1.199 g/mL, 1.215 g/mL, and 1.243 g/mL, respectively) were prepared by diluting a 60% Opti-prep™ stock solution (60% Iodixanol; Sigma-Aldrich Chemicals, Zwijndrecht, The Netherlands) with PBS. EVs in 45% Opti-prep™ were layered at the bottom of the ultracentrifuge tube and overlaid with the layers of 40%, 35%, 30%, 25% and 20% solutions. The gradient was ultracentrifuged for 18–24 h at $103,000 \times g$ at 15 °C (Beckman SW 60 Ti rotor). After ultracentrifugation, fractions of 0.5 mL were collected from the top of the tube. A portion of each fraction was precipitated with TCA and visualized by SDS-PAGE with the Pierce Silver Stain Kit (Thermo Fisher Scientific) or Coomassie staining according to the manufacturer's instructions. Fractions were also visualized by TEM. EVs were present in the region of the gradient corresponding to fractions 30–35%. Fractions containing EVs were pooled, diluted in ~ 12 mL of PBS and pelleted by ultracentrifugation ($220,000 \times g$, 3 h, 15 °C, Beckman SW41 rotor). Purified EVs were resuspended in PBS 1X buffer.

## EV quantification and characterization

**NanoFCM.** The EVs concentration and size distribution was determined by flow cytometry (NanoFCM, Inc., Xiamen, China). The instrument was aligned, focused, and calibrated using 0.25 μm Fluorescent Silica Microspheres, that also served as a concentration standard. Silica Nanosphere Cocktail (S16M-Exo) was used as a size standard (NanoFCM, Inc.; range 68–155 nm), from which a calibration curve was calculated and used to infer the size of the events present in each sample using the NanoFCM software (NanoFCM Profession V2.0). All samples were diluted with 0.02 μm filtered PBS 1X solution to ensure the particle count was within the range of 2000–12,000/min. All particles that passed by the detector over 60 s intervals were recorded.

**CytoFLEX nano.** The EVs sample was also analyzed by a flow cytometer CytoFLEX nano (Beckman Colter, Brea, CA, USA). The EVs preparation was diluted in filtered 1 × PBS. The fluidic system ran at a constant speed of 3 μL/min, and the sample acquisition lasted 1 min. Nanosphere™ series 3000 (Thermo Scientific) polystyrene beads with standard sizes of 80, 125, 200, 300, and 450 nm were used for light scatter calibration using FCMPass software v4.1.1[72]. The 405 nm laser light was used for the detection of side scatter (SSC).

**Nanoparticle tracking analysis (NTA).** The size distribution of EVs was analyzed by NTA using NanoSight Pro (Malvern Panalytical, UK). Samples were diluted in 1 x PBS according to the manufacturer's instructions (final concentration between 107 - 109 particles per ml). A 488 nm laser was used for data acquisition. Captures of 23.1 s (750 frames) were performed in five replicates with a cell temperature of 25 °C and a flow rate of 3 μl/min. Data

output was acquired using NanoSight NTA software version 1.1.0.6 (Malvern Instruments).

## Transmission electron microscopy (TEM)

Before staining, carbon-coated copper grids were incubated with a 0.1% of poly-L-lysine solution in $H_2O$ for 30 min[73]. Coated grids were washed three times with water, and subsequently, 5 μL of the sample was applied to the grid and negatively stained using 2% uranyl acetate (wt/vol). Samples were imaged with the transmission electron microscope FEI Spirit Tecnai Biotwin operating at 120 kV. The relative diameter of EVs was determined as previously described[19]. Briefly, electron micrographs of negatively stained EVs were analyzed with ImageJ[74], and the area of each was calculated manually ($n = 412$). The relative diameter was calculated according to the equation $A = (\pi/4) \times D^2$.

## DNA isolation from EVs and sequencing

Before DNA extraction, EVs were treated with DNase I (15 μ/ml) and RNase (100 μg/ml) in the presence of $MgCl_2$ for 30 min at 37 °C, followed by the addition of EDTA (20 mM). For DNA extraction, *M. smithii* EVs were treated with proteinase K (100 μg/ml) and SDS (0.5%) at 55 °C for 30 min. The DNA was extracted by the standard phenol/chloroform/isoamyl alcohol (25:24:1 vol/vol) procedure[19]. The digestion of the extracellular DNA not protected by EVs was assessed using fluorescence microscopy of DAPI-stained EV preparation. Subsequently, EVs were disrupted by incubation with SDS and proteinase K at final concentrations of 0.5% (wt/vol) and 100 μg/ml, respectively, for 30 min at 55 °C. The DNA was extracted with the mixture of phenol/chloroform/isoamyl alcohol (25:24:1 vol/vol/vol) and precipitated with sodium acetate to a final concentration of 0.3 M and ice-cold 70% ethanol. Sequencing libraries were prepared and sequenced on the Illumina MiSeq platform with 150-bp paired-end read lengths (Institut Pasteur, France). Raw sequence reads were processed with Trimmomatic v.0.3.6[75] and mapped to the reference genome of *M. smithii* PS using Bowtie2 with default parameters[76] and visualized with UGENE[77]. In addition, raw sequences were assembled with MetaSPAdes v3.11.1 with default parameters[78]. For the contig corresponding to the extrachromosomal circular element, open reading frames (ORFs) were predicted by Prokka v.1.14.5[79]. Searches for distant homologs were performed using HHpred against PFAM, PDB and CDD databases[80].

## Mass spectrometry and data analysis

The protein content of *M. smithii* PS cells and purified EVs (triplicates) were analyzed by liquid chromatography – tandem mass spectrometry (LC-MS/MS) at the Proteomics Platform of Institut Pasteur (Paris, France) as previously described[19]. Briefly, samples were snap-frozen in liquid nitrogen, lyophilized and re-suspended in 100 μl of lysis buffer including 8 M Guanidine HCl (GuHCl), 5 mM Tris(2-carboxyethyl) phosphine (TCEP). Samples were sonicated in a Covaris E220 (Covaris) for 5 min at 200 cycles/burst, with 175 W peak power and a 10% duty cycle. Lysates were centrifuged 15 min, $15,000 \times g$ at RT to and supernatants were kept. 2-chloro-acetamide (CAA) was added to a final concentration of 20 mM. Subsequently, samples were incubated at 95 °C for 5 min, and 9 times volume samples of 50 mM Tris-HCl (pH 8.0) were added to dilute GuHCl to a concentration of under 1 M. A mixture of 2 μg of Trypsin/Lys-C (Promega – V5071) was added to the samples and kept at 37 °C overnight for the digestion of the proteins. The reaction was stopped by adding formic acid (FA) at 1% final. Peptides were desalted using Sep-Pac C18 Cartridges (Waters, USA) and eluted with 80% ACN / 0.1% FA. The purified peptides were concentrated to near dryness, re-suspended in 25 μl of 2% ACN / 0.1 % FA and analyzed by Nano LC-MS/MS using an EASY-nLC 1200 system (peptides were loaded and separated on a 50 cm long home-made C18 column; 75 μm ID, 1.9 μm particles, 100 Å pore size, ReproSil-Pur Basic C18 - Dr. Maisch GmbH, Ammerbuch-Entringen, Germany, coupled to

an Orbitrap Lumos tribrid (Thermo Fisher Scientific) tuned to the DDA mode). Peptides were eluted with a multi-step gradient from 5 to 25% buffer B (ACN 80% / FA 0.1%) in 95 min, 25 to 40% buffer B in 15 min and 40 to 95% Buffer B in 10 min at a flow rate of 250 nL/min for up to 130 min. Column temperature was set to 60 °C.

Mass spectra were acquired using Xcalibur software using a data-dependent Top 2 s method with a survey scans (300–1700 m/z) at a resolution of 60,000 and MS/MS scans (fixed first mass 110 m/z) at a resolution of 15,000. The AGC target and maximum injection time for the survey scans and the MS/MS scans were set to 6.0E + 05, 50 ms and 5.0E + 04, 100 ms, respectively. The isolation window was set to 1.6 m/z and normalized collision energy fixed to 30 for HCD fragmentation. We used a minimum intensity threshold of 5E + 04. Precursor ion charge states from 2 to 7 were accepted, and advanced peak determination was enabled. Exclude isotopes was enabled, and selected ions were dynamically excluded for 45 s.

Peptide masses were searched against a UniProt *M. smithii* PS database (1783 entries the 19/01/2023) using Andromeda[81] with the MaxQuant ver. 2.0.3.0[82] software. Variable modifications (methionine oxidation and N-terminal acetylation) and fixed modification (cysteine carbamidomethylation) were set for the search, and trypsin with a maximum of two missed cleavages was chosen for searching. The minimum peptide length was set to 7 amino acids, and the false discovery rate (FDR) for peptide and protein identification was set to 0.01. The main search peptide tolerance was set to 4.5 ppm and to 20 ppm for the MS/MS match tolerance. The second peptides were enabled to identify co-fragmentation events. Identified proteins were functionally annotated using the archaeal clusters of orthologous groups (arCOG) database[44]. Differential expression of proteins was calculated using the R package DEP (differential enrichment analysis of proteomics data) (v. 1.21.0) as described in refs. 15,83. Data underwent variance stabilizing transformation for normalization with the vsn function in the DEP package. The threshold for significant enrichment in EVs compared to cells is $\log_2$ fold change greater than 1.5 and adjusted *P*-value lower than 0.05.

**Detection of the extrachromosomal circular element by PCR**
Polymerase chain reactions (PCRs) with primers targeting the integrated (F: 5′-TCTTCAGGACTTACATCCAGG-3′, R: 5′-TGTACGTTCACATCCGT CTA-3′; expected size: 291 bp) and excised (F: 5′-CTGTTGAAGAAG GTAAACCCG-3′, R: 5′-TTGTACGTTCACATCCGTCT-3′; expected size: 433 bp) form of the extrachromosomal circular element were performed both on *M. smithii* cells and purified EVs. PCRs were performed using DreamTaq DNA Polymerase (Thermo Fisher Scientific) with the following steps: Denaturation at 95 °C for 3 min followed by 33 cycles of 95 °C 30 s, 57 °C 30 s, 72 °C 1 min, and a final extension step at 72 °C for 10 min. The sequences of the amplified products were confirmed by Sanger sequencing and analyzed with QIAGEN CLC Main Workbench 24.0 (QIAGEN, Aarhus, Denmark).

**Detection of MSTV1 by PCR**
PCRs with primers targeting the integrated (F: 5′-GGGTTTAA TTTTGGGGGATA-3′, R: 5′-AGGATTTCTTCATTGGTTCTCA-3′) and excised (F: 5′-TTGATGATGTTAATAATGGTGATGA-3′, R: 5′- AGGATTT CTTCATTGGTTCTTCTCA-3′) forms of MSTV1 were performed both on *M. smithii* cells and purified EVs[43]. PCRs were performed using DreamTaq DNA Polymerase (Thermo Fisher Scientific) with the following steps: denaturation at 95 °C for 3 min followed by 35 cycles of 95 °C 30 s, 57 °C 30 s, 72 °C 1 min, and a final extension step at 72 °C for 10 min.

**Cryo-ET: Sample preparation and tilt series acquisition**
Samples for cryo-electron tomography were prepared as described previously[43]. Briefly, a solution of bovine serum albumin–gold tracer containing 10-nm-diameter colloidal gold particles was added to a fresh culture of *M. smithii* in its exponential phase with a final ratio of 1:1. A small amount of the sample was applied to the glow-discharged (ELMO, Corduan) carbon-coated copper grids (Cu 200 mesh Quantifoil R2/2). The sample was rapidly frozen in liquid ethane using a Leica EMGP system. The grids were stored in liquid nitrogen until image acquisition. Tilt series were collected on a 300 kV Titan Krios G3 transmission electron microscope (Thermo Fisher Scientific) equipped with a X-FEG Tip, a Gatan K3 Direct Electron Detector and a Gatan BioQuantum LS Imaging Filter with slit width of 20 eV and a single-tilt axis holder. Tilt series were acquired with Tomography software v.5.6 (Thermo Fisher Scientific) using a dose-symmetric scheme[84], with an angular range of ± 60°, 2° angular increment, -8 μm defocus, pixel size of 3.4 Å (26000x). The total dose was set at 140 e/Å² at a dose rate of 41.6 e/pix/sec in vacuum, with C2 and Objectif Apertures of 100 μm. 3D tomographic reconstructions were calculated in IMOD by weighted back projection using the SIRT-like filter with 9 iterations[85].

**Cryo-ET: Segmentation and analysis of tomographic data**
The drawing tools of IMOD[85] were used for tomogram annotation. Archaeal membrane and cell wall were manually traced every 30 slides, and the subsequent use of the interpolator tool. Both closed and open contours were employed, depending on whether the full cell was displayed in the field of view. EVs were modeled by manual tracing in all the slices where they were present. All traces were merged through the "merge" tool. Surfaces were generated using the 'imodmesh' function. The IMOD surfaces were then imported to UCSF ChimeraX[86] together with the tomographic file. IMOD surfaces were either used directly in visualization or used to mask out regions of the tomographic volume with the 'volume mask' tool of ChimeraX. The subtomographic areas of interest were then visualized in iso-surface representations of variant threshold values and colors.

**Reporting summary**
Further information on research design is available in the Nature Portfolio Reporting Summary linked to this article.

## Data availability
The mass spectrometry proteomics data generated in this study have been deposited in the ProteomeXchange Consortium via the PRIDE partner repository with the dataset identifier PXD053033. Source data are provided in this paper.

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

## Acknowledgements

This work was supported by Agence Nationale de la Recherche (grant ANR-23-CE02-0022 to S.G. and M.K., and grant ANR-22-CE02-0003 to V.C.-K.). C.M.G. was supported by an FRM *Retour en France* fellowship. N.P. was supported by a Pasteur-Roux Postdoctoral Fellowship from the Institut Pasteur (Paris) and the Austrian Science Fund (FWF) Elise Richter Fellowship (FWF project V 931-B). We acknowledge the cryo-ET expertise and assistance of the Institut Pasteur's NanoImaging Core facility, created and supported by a PIA grant (EquipEx CACSICE: ANR-11-EQPX-0008). We also acknowledge E. Turc and L. Lemée from the Biomics Platform, C2RT, Institut Pasteur, Paris, France, supported by France Génomique (ANR-10-INBS-09) and IBISA. We are grateful for the support of the Ultrastructural BioImaging Core Facility equipment from the GIS-IBISA, the French Government Program Investissements d'Avenir France BioImaging (FBI, N° ANR-10-INSB-04-01) and the French government (Agence Nationale de la Recherche) Investissement d'Avenir program, Laboratoire d'Excellence "Integrative Biology of Emerging Infectious Diseases" (ANR-10-LABX-62-IBEID). The authors are grateful to Samantha Bauchiero for analysis of the EVs sample on the Cytoflex nano flow cytometer (Beckman Colter, Brea, CA, USA) and Dr Aymeric Audfray for analysis of the EVs sample on NanoSight Pro (Malvern Panalytical, UK).

## Author contributions

M.K., S.G., G.B., and D.P.B. conceived the study; S.G. provided infrastructure to grow and manipulate archaeal methanogens; D.P.B. performed the experiments and data analyses; V.C.-K., P-H.C., and D.P.B. measured the EV size distributions; C.M.-G. conducted preliminary experiments on the interaction of EVs with gut bacteria; N.P., A.S.-R., and S.T. prepared the samples and collected the cryo-EM data; A.G. reconstructed the tomogram; T.D. and M.M. prepared the samples and performed mass-spectrometry. D.P.B. and M.K. wrote the manuscript, which was revised and approved by all co-authors.

## Competing interests

The authors declare no competing interests.
