## [Transparent Peer Review file · Nature Communications]

Biogenesis of DNA-carrying extracellular vesicles by the dominant human gut methanogenic archaeon

Corresponding Author: Dr Mart Krupovic

Version 0:

Reviewer comments:

Reviewer #1

(Remarks to the Author)

In this work, the authors characterized the composition and biogenesis of EVs released by the dominant human gut methanogen, *M. smithii*, which possess a peptidoglycan cell wall. The EVs contain eccDNA and proviral DNA. Cryo-ET analysis showed that the small-diameter EVs are trapped in the pocket between the cytoplasmic membrane and the cell wall. In addition, the EVs contain proteins involved in methanogenesis with potential implications for methane production in the human gut. The experiments are well designed and the results are well organized.

This is the first report to characterize EVs from human gut methanogenic archaea. On the other hand, there are not enough novel and surprising results compared to previous reports on EV production. I am concerned that specific characteristics of methanogenic archaea in the human gut were not observed and detailed analyses are needed. Overall, based on novelty and significance, it is preferable to publish in a microbiology journal rather than a general journal such as *Nat Commun*. My specific comments are as follows.

Major comments

1. The proposed models of EV formation are shown in Figure 4, but the detailed mechanism has not been shown in this study. This model should not be presented because it has not been proven that eccDNA is encapsulated in EVs due to budding/blebbing, and that proviral DNA is contained in EVs derived from explosive cells. If these two types of EVs are different in size and density, they can be separated.
2. The authors showed that eccDNA and MSTV1 provirus are contained in EVs, but this event would be because their copy numbers are more than chromosomal DNA and the small size circular DNA can be encapsulated in EVs. Although the authors suggest that DNA-containing EVs would be carriers of HGT, the detailed functions of *M. smithii* EVs are still unknown.
3. EV formation by explosive cell lysis by prophage has been shown in several microbes, but there is no evidence for explosive cell lysis-mediated EV production in this archaea. The authors should at least provide data suggesting that EVs result from explosive lysis.
4. The accumulation of vesicles in the periplasmic spaces has been observed in Gram-negative bacteria (Takaki et al 2020 *Appl Environ Microbiol* e01131-20; Ojima et al 2021 *Front Microbiol* 12:706525) and vesicle production occurs through the peptidoglycan pole in Gram-positive bacteria (Toyofuku et al. 2017 *Nat Commun* 8:481). Based on the above reports, the EV production model presented in this study is very similar to them. It is necessary to clearly indicate what the *M. smithii*-specific EV biogenesis is.
5. The authors emphasize that it is the EV of gut archaea, but they do not mention what is the unique phenomenon of gut archaea, since its production is a mechanism that can also occur in other archaea or bacteria. There is a need for data that would broaden the discussion, such as how it relates to gut epithelial cells.

Other comments

6. Since the violin plot (Fig. 1C) is plotted from the TEM images (based on the text on page 14, lines 613-614), the authors need to show the calculation method and describe in the legend that the data were obtained from the TEM images.

7. EVs and the periplasmic showed no apparent presence of peptidoglycan coat in the TEM and cryo-ET images (page 3, line 126-127; page 6, line 282-283). The authors should address how the presence or absence of the cell wall component is detected.
8. In Fig. S2, is there a result of the sequence analysis of the *M. smithii* PS chromosome around 79634 bp? How could the 2.9 kbp DNA be the circularized form? Is there such a thing as an attachment site?
9. The legends seem to be misaligned, line 158 (Fig. 2C, Fig. S2-S4), line 191 (Fig. S5), line 206 (Fig. S6).
10. In Fig. 4A (i)(ii)(iii), could the authors please distinguish the EVs, cell wall and cytoplasmic membrane by arrows?

Reviewer #2

(Remarks to the Author)

Baquero et al have provided an insightful exploration of extracellular vesicles from *Methanobrevibacter smithii*, a dominant methanogen in the human gut. By employing a combined approach that includes NanoFCM, sequencing, electron microscopy, and mass spectrometry, the authors have investigated the biosynthesis and properties of EVs in the methanogens. This study advances our understanding of EV production in methanogens and beyond. The research is well-structured, and the data largely support the author's conclusions. A few questions and clarifications need to be addressed before publication.

Line 69: "monoderm (gram-positive) and diderm (gram-negative) bacteria "

Although monoderm bacteria tend to be Gram-positive, and diderm bacteria tend to be Gram-negative, the terms are not equivalent. Also, 'Gram' should be capitalized because it's named after a person.

Line 98: "pseudo-periplasmic space"

Please define the pseudo-periplasmic space here for a non-expert audience.

Line 106-113. These descriptions could be moved to the "Method section" with Figure 1A included as a supplementary Figure. If you choose to keep this in the Results section, please specify whether this is the first protocol for purifying EVs from methanogen and clarify how it differs from or improves upon existing purification protocols.

Line 124-125, Line 134, Line 183: Please clarify the threshold used to define the small and large EVs. It would be helpful to include percentage of small and large EVs within the analysed population.

Line 131-132. Was the treatment of DNase I to eliminate extracellular DNA successful? How was this determined? Please provide supporting evidence/figures.

Line 146. The figure citations are incorrect. They should refer to Figure S1-S3. Figure S4 is related to the detection of MSTV1. Please correct the following citations accordingly.

L157-158: "both playing a critical role in methanogenesis"

This is an overstatement, as is L31-32 of Abstract: "The eccDNA encodes two of the key methanogenesis enzymes and could boost their expression inside the cells through the gene dosage effect". As stated, the genes referred to here, Msm_0079 and Msm_0080, are involved in pathways responsible for the synthesis of two separate cofactors, F420 and coenzyme M, respectively. Neither protein is directly involved in methanogenesis. In fact, as the authors would be aware, certain methanogens (most notably *M. ruminantium*) do not synthesize coenzyme M.

L165: Please clarify whether the promoters are predicted or present upstream the Msm_0079 and Msm_0080, if they are, it would be helpful to label them on the map of circular element in Fig. 2D.

L166-168 "the additional gene copies from eccDNA must boost the expression of the corresponding genes and have an impact on methanogenesis."

This statement requires justification with regard to the inferred impact on methanogenesis. Does either enzyme (5-amino-6-(D-ribitylamino)uracil:L-tyrosine 4-hydroxybenzyltransferase and sulfopyruvate decarboxylase) catalyze a rate-limiting step in their respective biosynthetic pathways? How do the authors account for only the alpha subunit of sulfopyruvate decarboxylase (not the truncated beta subunit) having additional copies?

Line 178: The citation of Figure S5 should be corrected to Fig. S4.

L 184-186: "This hypothesis could not be confirmed experimentally using the *M. smithii* strains available in the laboratory, either due to inability of the EVs to overcome the cell wall barrier of the target cells or due to resistance mechanisms that remain to be understood. "

The inability of these virus-containing EVs to infect other *M. smithii* cells might suggest that the hypothesis is not correct.

L 192. The citation of Fig.S6 should be corrected to Fig. S5.

Line 193-195: What does the similar number of proteins found in *M. smithii* EVs, and *S. islandicus* EVs, and *H. lacusprofundi* EVs suggest? "lacusprofundi" should be corrected to "lacusprofundi".

Line 265-267: It would be helpful to label this local opening area in the Figure 4A.

Line 280-281: "We show that *M. smithii* EVs carry three types of DNA molecules which vastly differ in their abundancies."

A minor point - is "abundancies" a typo for "abundances"? Please clarify how the abundance of the three types of DNA molecules differs here.

Line L282-285 and L300-302: "These correspond to (i) random genomic loci distributed across the *M. smithii* chromosome, (ii) MSTV1 provirus and (iii) an extrachromosomal circular DNA (eccDNA) molecule of 2.9 kb. The frequency of incorporation into EVs likely depends on the availability (copy number) and dimensions of the corresponding molecules in the cell during the EV biogenesis."

"However, the presence of the provirus and eccDNA in the extrachromosomal form increases their copy number and hence the probability of being incorporated into EVs, partly explaining their higher abundance in the EVs."

This might suggest that the incorporation of *Msm_0078* to *Msm_0080* into a higher-copy number EV might not be due to the functional significance of these genes for methanogenesis. Because the mechanism responsible for this circular extrachromosomal element is not known (also stated on L335), then how can the authors rule out the possibility that the selection of this segment of the chromosome that includes *Msm_0078* to *Msm_0080* is due to the function of these three genes?

Line 295-296. Could this also be related to the size of EVs with different genomic fragments?

Lastly, it would be great helpful if authors can provide some new insights based on this research, such as a predicted model of evolution of extracellular vesicles across the evolutionary spectrum, and potential biotechnological roles of methanogen EV.

Reviewer #3

(Remarks to the Author)

Baquero and colleagues discuss the biogenesis and composition of extracellular vesicles produced by *Methanobrevibacter smithii*. The authors have analyzed the quantity, size and composition of these EVs and speculate a range of functions that may be mediated by these EVs based on their composition. Specifically, the authors report that *M. smithii* EVs contain extrachromosomal circular DNA encoding for methanogenesis enzymes that they suggest may boost their expression inside the cell. In addition, they propose a mechanism whereby *M. smithii* EVs are formed. The methods used for the isolation of EVs in this study are thorough, and care has been taken to remove contaminants from EV samples prior to subsequent analyses to validate their results.

Overall comments:

Collectively, the findings reported in this study are a descriptive analysis of *M. smithii* EVs, specifically focusing on the examination of (i) the DNA associated with DNase-treated *M. smithii* EVs, (ii) the proteome of *M. smithii* EVs compared to *M. smithii* bacteria and (iii) visualizing the production of EVs using CryoEM. Although these findings are interesting, this very descriptive study detailing the characteristics and composition of *M. smithii* EVs is complementary and possibly additive to a larger study performed by Baquero et al (under review). Furthermore, most of the findings reported in this study are based on characterization of the composition or morphology of these EVs without any mechanistic evidence, and therefore there are numerous instances where the authors overinterpret their findings without sufficient experimental evidence to support their hypotheses. For example, the suggestion that EVs contain methanogenesis enzymes that "could boost their expression inside the cells through a gene dosage effect" is speculative and has not been proven by the authors. Furthermore, it is suggested that larger EVs contain the viral genome and therefore may spread virus to other cells, but the presence of viral DNA or virus in these larger EVs has not been examined in this study. Overall, this is a very descriptive study of the composition and morphology of *M. smithii* EVs, and one that relies heavily on data that is already under review, and that is referred to numerous times throughout this manuscript.

Specific comments:

- 1) As NanoFCM was used to size EVs, and that many of the EVs isolated were below the detection limit of the instrument, an additional nanoparticle counter should be used to accurately determine the size of EVs. I appreciate that TEM was also used to examine EVs, however the dehydration methods used to prepare EV samples for TEM analysis impacts the ability to determine their size, which is often underestimated. An additional nanoparticle sizing method should be used to accurately determine EV size, using an instrument that can accurately determine small EVs (<40nm).
- 2) A range of genes were identified within EVs, and the authors speculate their ability to boost expression of these genes and their ability to impact methanogenesis (Line 177). Do the authors have any data that suggests this?
- 3) The finding that *M. smithii* EVs contain viral DNA is not surprising considering that the authors have demonstrated that provirus MSTV1 is a mechanism whereby EVs are produced (under review).
- 4) Line 192- The authors propose that larger EVs may carry the viral genome and facilitate virus spread to non-infected cells.

The authors should separate small and large EVs using density gradient separation and extract DNA from larger EVs to confirm that larger EVs have the complete viral genome within them as suggested. This can be performed using their established EV gradient isolation method.

5) Line 228- the authors report that there is only one provirus protein found in all 3 replicates of the EV proteome analyses. Do the authors suggest that this protein is found within the EVs, or on the surface of EVs as part of the EV-corona? Why would only one viral protein be selected for incorporation or association with EVs? Is this selective cargo packaging into EVs, or protein association on the surface of EVs.

6) Line 230- Have the authors performed a gradient purification of the virus using the same gradient conditions to isolate their EVs to know at what fraction/density the virus will be located to support their comment?

7) Have the authors grown *M. smithii* in conditions that may induce endolysin activity, ie UV, stress etc as per other bacteria to observe if endolysin EV production can occur, ie possibly due to PeiW-like endolysin?

8) Line 319- the authors speculate that the size of the EVs is likely to be dictated by the diameter of the openings in the peptidoglycan cell wall, and that the median diameter of EVs was 32nm. How do the authors speculate that larger EVs are made? Do they propose that these larger EVs are viral induced? It seems that the authors suggest this may be the case (line 344). Evidence should be provided of this if it is the case.

9) Line 367- Why would EVs be released and then reintroduced back into the same cell? What evidence exists for this hypothesis to be proposed? It is not clear how this will finetune methanogenesis as proposed (line 369).

10) Figure 1 D- can the authors provide details as to the number of biological replicates represented in the NanoFCM histogram image.

Reviewer #4

(Remarks to the Author)

POINT-BY-POINT RESPONSES TO THE REVIEWERS' COMMENTS

Reviewer #1 (Remarks to the Author):

In this work, the authors characterized the composition and biogenesis of EVs released by the dominant human gut methanogen, *M. smithii*, which possess a peptidoglycan cell wall. The EVs contain eccDNA and proviral DNA. Cryo-ET analysis showed that the small-diameter EVs are trapped in the pocket between the cytoplasmic membrane and the cell wall. In addition, the EVs contain proteins involved in methanogenesis with potential implications for methane production in the human gut. The experiments are well designed and the results are well organized.

This is the first report to characterize EVs from human gut methanogenic archaea. On the other hand, there are not enough novel and surprising results compared to previous reports on EV production. I am concerned that specific characteristics of methanogenic archaea in the human gut were not observed and detailed analyses are needed. Overall, based on novelty and significance, it is preferable to publish in a microbiology journal rather than a general journal such as *Nat Commun*. My specific comments are as follows.

RESPONSE: We thank the reviewer for the remarks and valuable suggestions. However, we cannot agree that the manuscript lacks in novelty. Thus far, nothing has been published on EV biogenesis in peptidoglycan-containing archaea or archaea in the gastrointestinal tract of humans or other animals. Our data provides the first proteomic and genomic characterization of these EVs and provides insights into the EV biogenesis from cryo-ET data. It is clear from our data that the EVs of peptidoglycan-containing archaea are produced in a manner similar to that of bacteria, contrasting the eukaryotic-like budding-based EV production previously reported in other archaea.

We recognize that the novelty has not been clearly presented in the original version of the manuscript. Thus, we added the following comparison to other archaeal EVs in the Discussion: *“More generally, the mechanisms of EV biogenesis in archaea appear to be more complex and diverse than previously appreciated. There are at least three distinct EVs biogenesis pathways in archaea. In particular, EVs of thermoacidophilic archaea of the order Sulfolobales are generated by budding in an ESCRT-dependent manner²⁴; in hyperhalophilic archaea that lack the ESCRT system, EV budding depends on a small Ras-like GTPase²¹; by contrast, EVs of peptidoglycan-containing methanogenic archaea which lack both ESCRT machinery and the Ras-like GTPase appear to be produced by blebbing, likely due to turgor pressure. The first two pathways are similar to those operating in eukaryotes, whereas the third one resembles that reported in bacteria. Thus, biogenesis of archaeal EVs seems to occur through a combination of different mechanisms operating in the two other cellular domains and depends on the availability of molecular membrane remodeling machineries and architecture of the cell envelope.”*

In addition, we provided further information on what has been known about EV biogenesis in archaea in the Introduction: *“The molecular mechanisms of EV budding have been studied in thermoacidophilic and hyperhalophilic archaea, *S. islandicus* and *Haloferax volcanii*, respectively. In *S. islandicus*, similar to eukaryotes³⁸, EV budding is mediated by the ESCRT (endosomal complexes required for transport) system²⁴, a membrane remodeling machinery which plays a key role in cytokinesis³⁹⁻⁴¹. By contrast, in hyperhalophilic archaea which lack the ESCRT system, EV biogenesis was shown to depend on the small Ras-like GTPase²¹. Both ESCRT machinery components and the GTPase are strongly enriched in the corresponding EVs^{21,24}. Notably, some archaea, such as methanogens of the order Methanobacteriales, the dominant group of archaea in the animal gastrointestinal tract (GIT)³³, have a peptidoglycan polymer surrounding the cytoplasmic membrane and lack the S-layer^{31,34,35}. Whether gut methanogens with a rigid cell wall can produce EVs remains unknown.”*

We hope that these additions will help to further appreciate the novelty of our findings.

Major comments

1. The proposed models of EV formation are shown in Figure 4, but the detailed mechanism has not been shown in this study. This model should not be presented because it has not been proven that eccDNA is encapsulated in EVs due to budding/blebbing, and that proviral DNA is contained in EVs derived from explosive cells. If these two types of EVs are different in size and density, they can be separated.

RESPONSE: We concur with the reviewer that the detailed mechanisms on how the budding/blebbing occurs in archaea remains unclear. One may argue that detailed mechanisms of EV biogenesis are not fully understood not only for archaea but also for bacteria, despite extensive research in this latter area. This said, our cryo-ET data provides direct evidence of membrane deformation and presence of EVs trapped under the peptidoglycan, and shows that membrane blebs are protruding through the local openings in the peptidoglycan. Furthermore, we show that the eccDNA is present in EV preparations and is resistant to nuclease treatment, which is commonly interpreted as DNA being inside EVs rather than being freely present in the medium or adsorbed externally to the EVs. Thus, even though still hypothetical, the model shown in Figure 4B is not entirely baseless. We consider it appropriate and useful to present this model schematically in Fig. 4B and discuss it in the Discussion. Nevertheless, we added a question mark in the part relating to the explosive lysis and in the Discussion, we further emphasized that the model is a hypothesis that needs further experimental validation.

2. The authors showed that eccDNA and MSTV1 provirus are contained in EVs, but this event would be because their copy numbers are more than chromosomal DNA and the small size circular DNA can be encapsulated in EVs. Although the authors suggest that DNA-containing EVs would be carriers of HGT, the detailed functions of *M. smithii* EVs are still unknown.

RESPONSE: Indeed, we consider that enrichment of the MSTV1 genome and eccDNA in *M. smithii* EVs could, in part, be explained by their small size and increased copy number compared to the chromosomal DNA. This is what we stated in the Discussion of the original version of the manuscript: “*the presence of the provirus and eccDNA in the extrachromosomal form increases their copy number and hence the probability of being incorporated into EVs, partly explaining their higher abundance in the EVs*”.

Unfortunately, in the absence of genetic tools and selectable markers in *M. smithii*, we have no means to study the DNA transfer in this organism experimentally. In the case of genetically tractable hyperthermophilic archaea, we and others have shown that EVs can transfer recombinant plasmid DNA and defective virus genome from cell to cell (Gaudin et al., 2013 *Environ Microbiol Rep*, PMID: 23757139; Liu et al., 2021 *ISME J*, PMID: 33903726; Gaudin et al., 2014 *Environ Microbiol* PMID: 24034793). Therefore, the possibility that *M. smithii* EVs are also involved in intercellular transport of DNA cannot be dismissed. As mentioned in the manuscript, our attempts to demonstrate the infectivity of the MSTV1 DNA-carrying EVs were unsuccessful. More importantly, given that, unlike other archaea where EV-mediated DNA transfer was demonstrated, *M. smithii* cells are surrounded by a peptidoglycan layer, we consider the role of *M. smithii* EVs in intercellular DNA transfer to be unlikely (even if not impossible). Our hypothesis on this topic is summarized in the Discussion as follows:

“*However, given that Methanobacteriales cells are surrounded by a rigid cell wall, fusion of EVs with the cytoplasmic membrane might not be straightforward, if at all possible. It is thus more likely that EVs provide means for the discharge of damaged or viral DNA, or surplus components, both proteins and genes, that outlived their role under given conditions.*”

3. EV formation by explosive cell lysis by prophage has been shown in several microbes, but there is no evidence for explosive cell lysis-mediated EV production in this archaea. The authors should at least provide data suggesting that EVs result from explosive lysis.

RESPONSE: We agree with the reviewer that the presented evidence for the explosive lysis is not direct. We note that MSTV1, the provirus harbored by *M. smithii* PS, is chronically induced and resulting virus particles are constantly released from the cells through lysis at low frequency (Baquero et al., 2024 *Nat Commun*, PMID: 39231967). The virus replication could not be efficiently induced by any of the broad range of tested agents. Thus, we know that some cells undergo lysis, but due to the low frequency of these events and inability to induce the virus mediated lysis, we cannot observe the lysing cells directly. Based on the mass spectrometry results and the purification procedure used, we are confident that the DNase-resistant viral genomes found in the EV preparations are not encapsidated within virus particles. Thus, the viral genome-containing EVs could be produced either by blebbing through the peptidoglycan or through explosive lysis. Given that large diameter holes in the peptidoglycan and blebbing of large portions of the membrane would likely result in lysis, we consider the second of the two scenarios to be more parsimonious. Nevertheless, we toned down the claims about the explosive lysis and mentioned explicitly that this route is hypothetical. However, we prefer retaining the corresponding panel (with a question mark) in Figure 4B as a potential route for the biogenesis of larger EVs. The revised text reads as follows:

*“By contrast, the release of larger-diameter EVs carrying the viral genome (>100 nm) would necessitate large openings in the peptidoglycan cell wall that could have detrimental effects on cell integrity. Therefore, the mechanisms generating small and larger EVs could be distinct. Bacterial EVs carrying viral genomes are thought to be generated during explosive virus-mediated lysis. Although we did not observe the explosive lysis and production of larger EVs in cryo-ET, we hypothesize that the MSTV1 genome containing *M. smithii* EVs are generated through a similar process (Fig. 4B). However, this hypothesis remains to be validated experimentally.”*

4. The accumulation of vesicles in the periplasmic spaces has been observed in Gram-negative bacteria (Takaki et al 2020 *Appl Environ Microbiol* e011131-20; Ojima et al 2021 *Front Microbiol* 12:706525) and vesicle production occurs through the peptidoglycan pole in Gram-positive bacteria (Toyofuku et al. 2017 *Nat Commun* 8:481). Based on the above reports, the EV production model presented in this study is very similar to them. It is necessary to clearly indicate what the *M. smithii*-specific EV biogenesis is.

RESPONSE: We thank the Reviewer for pointing out the relevant references, some of which were not cited in the previous version of the manuscript (but now they are). We note that although EV biogenesis has been studied in different bacteria rather extensively, what we present in the current manuscript has never been reported in the domain Archaea. Indeed, all other archaea in which EV biogenesis has been studied to date do not contain the peptidoglycan, a key factor directly relevant for EV biogenesis. Prior to this study, it was unclear whether archaea with a rigid cell wall could produce EVs. Our results provide the first evidence that archaea with this distinctive peptidoglycan polymer are indeed capable of producing EVs. Notably, our findings do not support the budding mechanism proposed for other archaeal EVs. Instead, they point to an alternative model in which *M. smithii* EVs accumulate in the periplasmic space prior to release, resembling a mechanism described in certain bacteria, where EVs undergo a ‘waiting’ period until localized openings in the peptidoglycan polymer emerge. In the revised manuscript, we further emphasize the novelty of our results compared to the previous literature on archaeal EVs.

More generally, the mechanisms deduced in bacteria cannot be automatically assumed to be applicable to archaea until actual experiments in archaea are carried out. Accordingly, we

discuss our results in the context of bacterial EV models. These comparisons provide a powerful way to uncover similarities and differences between the two prokaryotic domains.

5. The authors emphasize that it is the EV of gut archaea, but they do not mention what is the unique phenomenon of gut archaea, since its production is a mechanism that can also occur in other archaea or bacteria. There is a need for data that would broaden the discussion, such as how it relates to gut epithelial cells.

RESPONSE: We would like to emphasize that our study focuses on the biochemical characterization and biogenesis of *M. smithii* EVs rather than the role of EVs in the gut. This said, methane production, exclusively performed by methanogenic archaea, is a process of global importance in the context of both carbon cycling and global warming. Our results show that *M. smithii* EVs are enriched in both DNA and proteins that appear to be directly relevant to methanogenesis. In particular, the small circular DNA molecules enriched in EVs encode some of the coenzymes that are critical for methanogenesis, whereas four of the top 10 most abundant EV proteins are related to methanogenesis.

Although the question of how *M. smithii* EVs specifically interact within the human intestinal environment is outside the scope of our current work, we note that this topic has been explored in a back-to-back submitted manuscript by Weinberger et al. (<https://doi.org/10.1101/2024.06.22.600174>). While our manuscript focuses on the characterization and biogenesis of *M. smithii* EVs, their study investigates the interactions between EVs from gut methanogenic archaea and host immune cells.

Other comments

6. Since the violin plot (Fig. 1C) is plotted from the TEM images (based on the text on page 14, lines 613-614), the authors need to show the calculation method and describe in the legend that the data were obtained from the TEM images.

RESPONSE: The calculation method is described in Material and Methods, section Transmission electron microscopy (TEM). We have modified the legend of Fig. 1D to specify that the data of the violin plot was obtained from TEM imaging.

7. EVs and the periplasmic showed no apparent presence of peptidoglycan coat in the TEM and cryo-ET images (page 3, line 126-127; page 6, line 282-283). The authors should address how the presence or absence of the cell wall component is detected.

RESPONSE: Thank you for this comment. It is not possible to determine based on the TEM analysis whether EVs have the peptidoglycan coat. Thus, we removed the former statement on page 3, line 126-127. However, analysis of juxtaposed cryo-ET images of EVs and cells, as shown in Figure 4A, clearly shows that whereas the cell envelope consists of two layers, peptidoglycan and membrane, EVs do not have the external peptidoglycan layer, with only the membrane being present. The membrane and peptidoglycan layer are now indicated with arrows in the revised Fig. 4A.

8. In Fig. S2, is there a result of the sequence analysis of the *M. smithii* PS chromosome around 79634 bp? How could the 2.9 kbp DNA be the circularized form? Is there such a thing as an attachment site?

RESPONSE: The sequencing depths of 15x across the junction of the integral chromosomal form and 9106x for the circularized form of the 2.9 kb region (Fig. 2A, 2B) that we obtained are fully sufficient to unambiguously state that the fragment exists in both chromosomal and extrachromosomal states. The “excision” was further confirmed by PCR and Sanger sequencing of the PCR product. The mechanism of excision remains unclear. We searched

the sequences bordering the 2.9 kb fragment for the presence of direct repeats, which could serve as “attachment sites” for homologous recombination, but failed to identify any.

9. The legends seem to be misaligned, line 158 (Fig. 2C, Fig. S2-S4), line 191 (Fig. S5), line 206 (Fig. S6).

RESPONSE: We apologize for this mistake. The legends of the supplementary figures are now corrected.

10. In Fig. 4A (i)(ii)(iii), could the authors please distinguish the EVs, cell wall and cytoplasmic membrane by arrows?

RESPONSE: We have added arrows to distinguish EVs, the peptidoglycan polymer, the cytoplasmic membrane and local openings in panels i, ii and iii of Figure 4.

Reviewer #2 (Remarks to the Author):

Baquero et al have provided an insightful exploration of extracellular vesicles from *Methanobrevibacter smithii*, a dominant methanogen in the human gut. By employing a combined approach that includes NanoFCM, sequencing, electron microscopy, and mass spectrometry, the authors have investigated the biosynthesis and properties of EVs in the methanogens. This study advances our understanding of EV production in methanogens and beyond. The research is well-structured, and the data largely support the author’s conclusions. A few questions and clarifications need to be addressed before publication.

RESPONSE: We thank the Reviewer for the positive comments on our manuscript and the valuable suggestions.

Line 69: "monoderm (gram-positive) and diderm (gram-negative) bacteria " Although monoderm bacteria tend to be Gram-positive, and diderm bacteria tend to be Gram-negative, the terms are not equivalent. Also, ‘Gram’ should be capitalized because it’s named after a person.

RESPONSE: Thank you for this remark. We corrected the sentence as follows: “The first mechanism is common to both monoderm (containing a single cytoplasmic membrane) and diderm (containing an inner and an outer membrane) bacteria”.

Line 98: “pseudo-periplasmic space” Please define the pseudo-periplasmic space here for a non-expert audience.

RESPONSE: We have replaced the term “pseudo-periplasmic space” with “periplasmic space” to maintain consistency in terminology throughout the text.

Line 106-113. These descriptions could be moved to the “Method section” with Figure 1A included as a supplementary Figure. If you choose to keep this in the Results section, please specify whether this is the first protocol for purifying EVs from methanogen and clarify how it differs from or improves upon existing purification protocols.

RESPONSE: We appreciate the Reviewer’s comment, but we would like to retain Figure 1A in the main text, as this is the first protocol describing the concentration and purification of EVs from peptidoglycan-containing archaea. This is now explicitly stated in the text: “In the absence of protocols for EV purification from peptidoglycan-containing archaea, we first established a protocol for obtaining high-purity EV preparations from *M. smithii*, a prerequisite for biochemical characterization (Fig. 1A)”.

Line 124-125, Line 134, Line 183: Please clarify the threshold used to define the small and large EVs. It would be helpful to include percentage of small and large EVs within the analysed population.

RESPONSE: We have now quantified the number of vesicles with diameters ranging from 45-100 nm (~ 97%) and those larger than >100 (~3%). Given that the detection limit of NanoFCM is ~40 nm, we cannot quantify accurately the number of small EVs (< 45 nm) that were detected using TEM.

Line 131-132. Was the treatment of DNase I to eliminate extracellular DNA successful? How was this determined? Please provide supporting evidence/figures.

RESPONSE: Following the purification procedure, we did not expect the EV preparations to be heavily contaminated with extracellular DNA and thus DNase I treatment was done primarily as a precaution. The efficiency of DNase I treatment was assessed using fluorescence microscopy. The EV preparation was stained with DAPI and visualized by fluorescence microscopy before and after DNase I treatment. The fluorescence observed prior to the DNase I treatment, presumably, corresponding to free extracellular DNA, was no longer observed following DNase I treatment, suggesting complete removal of extracellular DNA. We used 15U/ml of DNase I which is sufficient to digest 15 µg/ml of plasmid DNA in 10 minutes. Thus, we are confident that all accessible DNA would be digested under the conditions used. These details are now mentioned in Materials and Methods.

Legend: Fluorescence micrographs of EV preparations prior (left) and after (right) DNase I treatment. Arrowheads point to DAPI-stained DNA.

Line 146. The figure citations are incorrect. They should refer to Figure S1-S3. Figure S4 is related to the detection of MSTV1. Please correct the following citations accordingly.

RESPONSE: We apologize for this mistake. Figure citations are now corrected.

L157-158: "both playing a critical role in methanogenesis"

This is an overstatement, as is L31-32 of Abstract: "The eccDNA encodes two of the key methanogenesis enzymes and could boost their expression inside the cells through the gene

dosage effect". As stated, the genes referred to here, Msm_0079 and Msm_0080, are involved in pathways responsible for the synthesis of two separate cofactors, F420 and coenzyme M, respectively. Neither protein is directly involved in methanogenesis. In fact, as the authors would be aware, certain methanogens (most notably *M. ruminantium*) do not synthesize coenzyme M.

RESPONSE: Indeed, certain methanogens cannot synthesis coenzyme M and thus must take it up from their environment, because this is an essential cofactor of methanogenesis. Given that *Methanobrevibacter smithii* is often the sole methanogen in the human gut and coenzyme M is not synthesized by gut bacteria (i.e., there is generally no external source of coenzyme M), *M. smithii* must synthesize this essential element. As for F420, to our knowledge, there is no known auxotrophy/import of this cofactor in CO₂-reducing methanogens (like *M. smithii*). Even if both enzymes are not directly part of the methanogenesis pathway, they are both essential for the functioning of this pathway in *M. smithii* (and more generally, essential for *M. smithii*, as it can only obtain energy through methanogenesis).

L165: Please clarify whether the promoters are predicted or present upstream the Msm_0079 and Msm_0080, if they are, it would be helpful to label them on the map of circular element in Fig. 2D.

RESPONSE: Indeed, the two genes are preceded by divergent promoters which are indicated in the figure with the broken arrows. Now this is also mentioned in the figure legend.

L166-168 "the additional gene copies from eccDNA must boost the expression of the corresponding genes and have an impact on methanogenesis.

This statement requires justification with regard to the inferred impact on methanogenesis. Does either enzyme (5-amino-6-(D-ribitylamino)uracil:L-tyrosine 4-hydroxybenzyltransferase and sulfopyruvate decarboxylase) catalyze a rate-limiting step in their respective biosynthetic pathways? How do the authors account for only the alpha subunit of sulfopyruvate decarboxylase (not the truncated beta subunit) having additional copies?

RESPONSE: In the absence of experimental evidence showing that the additional gene copies present in the eccDNA boost the expression of the corresponding genes, and following the recommendation of Reviewer 3, we have removed the statement. The revised sentence reads: "The presence of putative promoters in front of Msm_0079 and Msm_0080 suggests that the two genes could be transcribed."

Line 178: The citation of Figure S5 should be corrected to Fig. S4.

RESPONSE: We apologize for this mistake. Figure citations are now corrected.

L 184-186: "This hypothesis could not be confirmed experimentally using the *M. smithii* strains available in the laboratory, either due to inability of the EVs to overcome the cell wall barrier of the target cells or due to resistance mechanisms that remain to be understood. "

The inability of these virus-containing EVs to infect other *M. smithii* cells might suggest that the hypothesis is not correct.

RESPONSE: We agree with the Reviewer that it is possible, perhaps even likely, that EVs containing the MSTV1 genome are not involved in the virus spread in the population. However, as quipped by Carl Sagan, absence of evidence is not evidence of absence. In our recent paper on the characterization of MSTV1 (Baquero et al., 2024 *Nat Commun*, PMID: 39231967), we showed that the virus itself exhibits a narrow host range, being incapable of infecting the provirus-free *M. smithii* strains tested. Given that we do not have *M. smithii* strains sensitive to

MSTV1 infection, we cannot exclude the possibility that EVs could deliver the viral DNA to sensitive *M. smithii* strains that may exist in natural environments. Thus, we think it appropriate to mention this hypothesis in the text.

L 192. The citation of Fig.S6 should be corrected to Fig. S5.

RESPONSE: We apologize for this mistake. Figure citations are now corrected.

Line 193-195: What does the similar number of proteins found in *M. smithii* EVs, and *S. islandicus* EVs, and *H. lacusprofundi* EVs suggest? "*lacusprofundi*" should be corrected to "*lacusprofundi*".

RESPONSE: The comparison of the number of proteins identified in our study with those reported in other proteomic analyses of archaeal EVs provided confidence in the suitability of our EV purification protocol. More generally, the similar number of proteins found in EVs produced by distantly related archaea thriving in highly different environments also suggests the existence of a similar principle of cargo protein incorporation into EVs. Despite statistically significant enrichment in certain protein species in EVs when compared to the cellular proteomes, the majority of incorporated proteins represent those that are most abundant in the cells and are likely to be incorporated non-specifically. The same is likely to be true for EVs produced by other archaeal species, but this hypothesis remains to be tested once EVs from a broader range of archaea are quantitatively analyzed by mass spectrometry.

The typo in "*lacusprofundi*" has been corrected. Thank you for pointing it out.

Line 265-267: It would be helpful to label this local opening area in the Figure 4A.

RESPONSE: We thank the Reviewer for this suggestion and have modified Figure 4A accordingly. The opening is now indicated with arrowheads.

Line 280-281: "We show that *M. smithii* EVs carry three types of DNA molecules which vastly differ in their abundancies."

A minor point - is "abundancies" a typo for "abundances"? Please clarify how the abundance of the three types of DNA molecules differs here.

RESPONSE: Indeed, it was an unfortunate typo – now corrected. In the revised version, we included in parentheses the average sequencing depths for each type of DNA molecule found in the EVs.

Line L282-285 and L300-302: "These correspond to (i) random genomic loci distributed across the *M. smithii* chromosome, (ii) MSTV1 provirus and (iii) an extrachromosomal circular DNA (eccDNA) molecule of 2.9 kb. The frequency of incorporation into EVs likely depends on the availability (copy number) and dimensions of the corresponding molecules in the cell during the EV biogenesis. "

"However, the presence of the provirus and eccDNA in the extrachromosomal form increases their copy number and hence the probability of being incorporated into EVs, partly explaining their higher abundance in the EVs. "

This might suggest that the incorporation of Msm_0078 to Msm_0080 into a higher-copy number EV might not be due to the functional significance of these genes for methanogenesis. Because the mechanism responsible for this circular extrachromosomal element is not known (also stated on L335), then how can the authors rule out the possibility that the selection of this segment of the chromosome that includes Msm_0078 to Msm_0080 is due to the function of these three genes?

RESPONSE: Thank you for pointing out this possibility. Indeed, we cannot formally rule out that the 2.9-kb segment is specifically incorporated into EVs due to the importance of the encompassed genes. In the revised manuscript, we mentioned this alternative possibility in the Discussion:

“Although we did not detect sequence features, such as direct repeats, which would suggest site-specific excision of the 2.9-kb eccDNA segment, we cannot rule out the possibility that this chromosomal region is preferentially incorporated into EVs due to the importance of the encompassed genes for the cellular fitness.”

Line 295-296. Could this also be related to the size of EVs with different genomic fragments?

RESPONSE: Indeed, we propose that small EVs (~30 nm) can carry random chromosomal fragments and eccDNA, whereas the larger vesicles (>100 nm) are likely the ones enclosing the 38-kb viral genome. This is discussed in the text as follows: “Note that the diameter of the MSTV1 capsid in which the 38-kb viral genome is packaged under pressure is ~65 nm⁴⁹. Thus, although histone proteins detected in the EVs are likely to condense the cargo DNA, the volume and hence the diameter of the EVs enclosing the viral DNA is still likely to exceed 65 nm. NanoFCM analysis revealed a minor fraction (3.3%) of larger-diameter EVs ranging from 100 and 180 nm, which could be sufficient for carrying the full viral genome.”

Lastly, it would be great helpful if authors can provide some new insights based on this research, such as a predicted model of evolution of extracellular vesicles across the evolutionary spectrum, and potential biotechnological roles of methanogen EV.

RESPONSE: We appreciate the Reviewer’s comment and attempted to provide some insights into the potential role and applications of *M. smithii* EVs in the revised manuscript:

“It is increasingly recognized that gut bacteria-derived EVs modulate the intestinal microenvironment and play an important role in the gut homeostasis³⁷. While the role of archaeal EVs in the human gut remains unclear, the fact that *M. smithii* EVs transport both DNA and proteins suggests that they may act as messengers or regulators in archaea-archaea, archaea-bacteria and archaea-host interactions. Our results could lay a foundation for harnessing the EVs produced by *M. smithii* for developing drug and vaccine delivery systems in the human gut.”

We also added the following general evolutionary considerations:

“More generally, the mechanisms of EV biogenesis in archaea appear to be more complex and diverse than previously appreciated. There are at least three distinct EVs biogenesis pathways in archaea. In particular, EVs of thermoacidophilic archaea of the order Sulfolobales are generated by budding in an ESCRT-dependent manner²⁴; in hyperhalophilic archaea that lack the ESCRT system, EV budding depends on a small Ras-like GTPase²¹; by contrast, EVs of peptidoglycan-containing methanogenic archaea which lack both ESCRT machinery and the Ras-like GTPase appear to be produced by blebbing, likely due to turgor pressure. The first two pathways are similar to those operating in eukaryotes, whereas the third one resembles a vesiculation mechanism reported in bacteria. Thus, biogenesis of archaeal EVs seems to occur through a combination of different mechanisms operating in the two other domains and depends on the availability of molecular membrane remodeling machineries and architecture of the cell envelope.”

Reviewer #3 (Remarks to the Author):

Baquero and colleagues discuss the biogenesis and composition of extracellular vesicles

produced by *Methanobrevibacter smithii*. The authors have analyzed the quantity, size and composition of these EVs and speculate a range of functions that may be mediated by these EVs based on their composition. Specifically, the authors report that *M. smithii* EVs contain extrachromosomal circular DNA encoding for methanogenesis enzymes that they suggest may boost their expression inside the cell. In addition, they propose a mechanism whereby *M. smithii* EVs are formed. The methods used for the isolation of EVs in this study are thorough, and care has been taken to remove contaminants from EV samples prior to subsequent analyses to validate their results.

Overall comments:

Collectively, the findings reported in this study are a descriptive analysis of *M. smithii* EVs, specifically focusing on the examination of (i) the DNA associated with DNase-treated *M. smithii* EVs, (ii) the proteome of *M. smithii* EVs compared to *M. smithii* bacteria and (iii) visualizing the production of EVs using CryoEM. Although these findings are interesting, this very descriptive study detailing the characteristics and composition of *M. smithii* EVs is complementary and possibly additive to a larger study performed by Baquero et al (under review). Furthermore, most of the findings reported in this study are based on characterization of the composition or morphology of these EVs without any mechanistic evidence, and therefore there are numerous instances where the authors overinterpret their findings without sufficient experimental evidence to support their hypotheses. For example, the suggestion that EVs contain methanogenesis enzymes that “could boost their expression inside the cells through a gene dosage effect” is speculative and has not been proven by the authors. Furthermore, it is suggested that larger EVs contain the viral genome and therefore may spread virus to other cells, but the presence of viral DNA or virus in these larger EVs has not been examined in this study. Overall, this is a very descriptive study of the composition and morphology of *M. smithii* EVs, and one that relies heavily on data that is already under review, and that is referred to numerous times throughout this manuscript.

RESPONSE: We thank the Reviewer for valuable remarks and suggestions.

First, we would like to clarify that the study mentioned by the Reviewer has now been published (Baquero et al., 2024 *Nat Commun*, PMID: 39231967). This published study focused on the characterization of the virus MSTV1 and its interactions with the host in vivo and in vitro. There is no mention of *M. smithii* EVs in the published article and thus there is no overlap between the published and the current studies other than that the archaeal strain used is the same. The identification of the MSTV1 genome in *M. smithii* EVs is exclusively reported in the current manuscript. In particular, we show by sequencing that the viral DNA is present in the highly purified EVs that do not contain any of the viral structural proteins. We conclude that the viral genome is accommodated within the larger EVs based on the fact that small EVs cannot physically accommodate the complete viral DNA.

The Reviewer indicates that, in certain instances, we overinterpret the findings. For this we apologize and in the revised version have toned down our hypothesis regarding the role of the enzymes encoded by the extrachromosomal circular DNA found in EVs, as suggested by Reviewer 2 and removed the phrase “could boost their expression inside the cells through a gene dosage effect” from the text. We also emphasize that the presented models of EV biogenesis are hypothetical.

We hope to have addressed the concerns raised by Reviewer 3 and adjusted the text to align more closely with our findings. Our results, for the first time, shed light on the EV biogenesis by a peptidoglycan-containing archaeon and suggest a route different from that used by other studied archaea, while revealing parallels with vesiculation in bacteria.

Specific comments:

1) As NanoFCM was used to size EVs, and that many of the EVs isolated were below the detection limit of the instrument, an additional nanoparticle counter should be used to accurately determine the size of EVs. I appreciate that TEM was also used to examine EVs, however the dehydration methods used to prepare EV samples for TEM analysis impacts the ability to determine their size, which is often underestimated. An additional nanoparticle sizing method should be used to accurately determine EV size, using an instrument that can accurately determine small EVs (<40nm).

RESPONSE: We thank the Reviewer for this comment and agree that negative staining may alter the dimensions of the visualized EVs due to the flattening of the biological material following dehydration. This would result in slightly overestimated diameters of the EVs. In the revised manuscript, in addition to the EV size determination by nanoFCM and TEM, we present the results obtained with CytoFLEX Nano (Beckman Coulter) and Nanoparticle Tracking Analyzer (NTA; NanoSight Pro, Malvern Panalytical) instruments (new Figure S2). Analysis of the EV preparation using CytoFLEX Nano showed that the majority (93.4%) of EVs had a diameter of ~46–86 nm. Note, however, that ~40 nm is the detection limit of the instrument. By contrast, the mean diameter of *M. smithii* EV determined using NTA was 98 nm, which is at odds with the estimates obtained by all other approaches (TEM and flow cytometry). However, we note that NTA, which is widely used for determining the size of EVs, is subject to sensitivity limits due to the strong decrease in the intensity of scattered light scaling with particle diameter, causing the scattered light of very small particles to disappear below the background noise. Thus, NTA has reduced sensitivity for EVs smaller than ~50 nm and results in an overestimation of EV sizes (Comfort et al., 2021 *J Vis Exp*, PMID: 33843938).

Figure S2. Size distribution of *M. smithii* EVs determined using flow cytometer (CytoFLEX Nano, Beckman Coulter) (A) and Nanoparticle Tracking Analysis (NanoSight Pro, Malvern Panalytical) (B) instruments.

To overcome the limitations mentioned above, we also determined the size of the EVs using cryo-ET, which allows the visualization of samples in their native state, without the use of staining or fixatives. As indicated in the cryo-ET section of the Results (lines 259-260), 31 particles were measured, with diameters ranging from 12 to 45 nm, with the median diameter of 22 nm. The smaller median diameter estimated by cryo-ET compared to that of negatively stained EVs is likely due to artifacts inherent to negative staining (e.g., dehydration). These results are now included in the revised manuscript text.

Figure. EV diameters determined by cryo-electron microscopy.

2) A range of genes were identified within EVs, and the authors speculate their ability to boost expression of these genes and their ability to impact methanogenesis (Line 177). Do the authors have any data that suggests this?

RESPONSE: Since we do not have experimental evidence showing that the additional gene copies present in the eccDNA boost the expression of the corresponding genes, and following the suggestions of Reviewers 2 and 3, we have removed this speculation from the text.

3) The finding that *M. smithii* EVs contain viral DNA is not surprising considering that the authors have demonstrated that provirus MSTV1 is a mechanism whereby EVs are produced (under review).

RESPONSE: We apologize for lack of clarity on this point. The study mentioned by the Reviewer (Baquero et al., 2024 *Nat Commun*, PMID: 39231967) characterizes the lifecycle of MSTV1 and provides insights into archaeal virus-host interaction in the human gut. The cited article shows that the MSTV1 virus particles are constantly produced by a small fraction of the population and are released into the medium without significantly impacting the population growth. However, in that work we did not show the EV production or consider their role in virus genome transfer. Only through the data obtained from the sequencing of *M. smithii* EVs in the current work (Figure 2) we could demonstrate that EVs carry the viral genome and hypothesize that they may act as a vehicle for the dissemination of the virus within the population.

4) Line 192- The authors propose that larger EVs may carry the viral genome and facilitate virus spread to non-infected cells. The authors should separate small and large EVs using density gradient separation and extract DNA from larger EVs to confirm that larger EVs have the complete viral genome within them as suggested. This can be performed using their established EV gradient isolation method.

RESPONSE: Thank you for this suggestion. We attempted to separate the larger and smaller EVs by rate zonal centrifugation, but the fraction of larger EVs is very small, as now quantified in the revised manuscript (EVs with diameters ranging from 100-200 nm account only for 3.3% of the EV population). This quantity of larger EVs is too small to be visible in the rate zonal

sucrose gradient, which generally yields diffuse bands. Our purification procedure relies on the density gradient which, given that smaller and larger EVs should have the same density, does not allow separating them. In the revised manuscript, we rephrased the sentence to state that the larger EVs are sufficiently large to accommodate the complete virus genome: “The larger EVs observed by nanoFCM (Fig. 1D) have sufficient internal volume to accommodate the complete viral genome”.

5) Line 228- the authors report that there is only one provirus protein found in all 3 replicates of the EV proteome analyses. Do the authors suggest that this protein is found within the EVs, or on the surface of EVs as part of the EV-corona? Why would only one viral protein be selected for incorporation or association with EVs? Is this selective cargo packaging into EVs, or protein association on the surface of EVs.

RESPONSE: The only viral protein which is incorporated into EVs corresponds to a homolog of the Orc1/Cdc6 AAA+ ATPase, which is suggested to be a key factor of a regulatory circuit controlling the switch between the temperate and lytic MSTV1 infection cycles. Particularly, we suggested that the viral Orc1/Cdc6 protein acts as an activator of the integrase transcription, thereby triggering virus genome excision (Baquero et al., 2024 *Nat Commun*, PMID: 39231967). The Orc1/Cdc6-like proteins inherently bind to DNA, hence we presume that this viral protein gets incorporated in EVs being bound to the viral DNA. This is now stated in the revised text.

6) Line 230- Have the authors performed a gradient purification of the virus using the same gradient conditions to isolate their EVs to know at what fraction/density the virus will be located to support their comment?

RESPONSE: As described in Baquero et al., 2024 *Nat Commun*, PMID: 39231967, MSTV1 is produced chronically in low amounts, and despite multiple efforts, we have not identified an inducer that could efficiently boost virus production to the levels required for running purification gradients. Our comment is based on the fact that MSTV1 is a non-enveloped tailed virus (related to tailed bacteriophages, class *Caudoviricetes*). EVs of variable diameters were predominantly present in the gradient fractions with the densities between 1.11 and 1.13 g/mL, whereas the density of tailed virus particles that lack lipids is known to be >1.2 g/mL (Eppley et al., 2022 *Proc Natl Acad Sci U S A*, PMID: 36256808). Furthermore, we did not observe any MSTV1 particles in the EV preparations by electron microscopy. This is now clarified in the revised manuscript.

7) Have the authors grown *M. smithii* in conditions that may induce endolysin activity, ie UV, stress etc as per other bacteria to observe if endolysin EV production can occur, ie possibly due to PeiW-like endolysin?

RESPONSE: We thank the Reviewer for this comment. Although it seems plausible that the viral PeiW-like endolysin could play a role in *M. smithii* EV biogenesis, as has been proposed for bacterial EVs, our proteomic data does not support this scenario. Neither the MSTV1 PeiW-like endolysin nor any other type of lysins were identified in the EV proteomics data.

Regarding the induction of the endolysin activity, we tested a broad panel of conditions (34 chemical, physical and biological agents) to induce the lytic replication of MSTV1. However, none of these conditions had strong impact on virus production (Baquero et al., 2024 *Nat Commun*, PMID: 39231967).

8) Line 319- the authors speculate that the size of the EVs is likely to be dictated by the diameter of the openings in the peptidoglycan cell wall, and that the median diameter of EVs was 32nm. How do the authors speculate that larger EVs are made? Do they propose that

these larger EVs are viral induced? It seems that the authors suggest this may be the case (line 344). Evidence should be provided of this if it is the case.

RESPONSE: Indeed, we consider it likely that the small (~30) and larger (>100 nm) EVs are produced through distinct mechanisms, with the small EVs being released by budding/blebbing (Fig. 4) and the larger ones through explosive virus mediated lysis. However, we agree with the Reviewer that the evidence for the latter process is insufficient. Given that the virus could not be efficiently induced by any of the tested agents, we have no means of testing the explosive lysis hypothesis. Thus, in the revised manuscript, we mention this possibility only in the Discussion and explicitly state that this EV biogenesis route is hypothetical.

9) Line 367- Why would EVs be released and then reintroduced back into the same cell? What evidence exists for this hypothesis to be proposed? It is not clear how this will finetune methanogenesis as proposed (line 369).

RESPONSE: Our cryo-ET data showed that EVs are trapped in the periplasmic 'pockets' between the cytoplasmic membrane with high local negative curvature and the peptidoglycan. Some of these EVs will escape through local lesions in the peptidoglycan (and those are the ones we harvest). However, given the spatial proximity of the two membranes (EV and cytoplasmic), there is no reason why the EVs could not fuse back with the cytoplasmic membrane. The energetic barrier for the fusion could be overcome either by certain proteins present on the either or both membranes or spontaneously under certain conditions, e.g., particular local lipid composition or high energy state membrane patches (e.g., with high curvature). Regardless of whether the periplasmic EVs fuse back with the cytoplasmic membrane or are released in the environment, the fact that some of the most abundant EV proteins are those directly implicated in methanogenesis suggests that their EV-mediated export (reversible or not) will regulate the methanogenesis process. We present this speculative, but in our opinion, likely possibility in the Discussion and hope that this is adequate.

10) Figure 1 D- can the authors provide details as to the number of biological replicates represented in the NanoFCM histogram image.

RESPONSE: We thank the Reviewer for this constructive suggestion. The size distribution was measured using NanoFCM four times with independent biological samples. This is now mentioned in the legend of Figure 1D. The individual replicates are shown in a new Supplementary figure 1.

Figure S1. Biological replicates of EV analysis using nanoFCM. Each replicate represents independently purified EV preparations.